# Problematic online gaming mediates the association between attention-deficit/ hyperactivity and subsequent mental health issues in adolescents

Zui C. Narita [1] ✉, Jordan DeVylder[2,3], Gemma Knowles[4], Shuntaro Ando [5], Syudo Yamasaki [3], Mitsuhiro Miyashita[3], Daniel Stanyon[4], Satoshi Yamaguchi[3], Toshiaki A. Furukawa [6], Kiyoto Kasai[5] & Atsushi Nishida[3]

Theoretical models suggest that attention-deficit/hyperactivity and problematic online gaming could contribute to negative mental health outcomes, yet evidence on their interplay remains limited. Using data from 3171 adolescents in the Tokyo Teen Cohort, the present study examined the interplay of attention-deficit/hyperactivity at age 12 and problematic online gaming at age 14 associated with mental health issues at age 16. The sample consisted of 1487 girls (46.9%) and 1684 boys (53.1%). Doubly robust estimation revealed that a high degree of problematic online gaming consistently elevated the absolute and relative risks of mental health issues. The fully adjusted risk difference and risk ratio (95% confidence intervals (CIs)) were: incident depression, 7.8% (3.0%–13.1%) and 1.62 (1.25–2.05); incident anxiety, 5.7% (2.7%–8.7%) and 1.98 (1.45–2.67); incident psychotic experiences, 5.9% (2.3%–10.8%) and 1.72 (1.30–2.47); diminished well-being, 9.6% (5.1%–14.3%) and 1.54 (1.27–1.84). Higher attention-deficit/hyperactivity scores were associated with a greater degree of problematic online gaming (adjusted $\beta$ per 1 SD: 0.18, 95% CI: 0.12–0.24). Causal mediation analysis—ensuring temporal plausibility, carefully adjusting for confounders, and accounting for exposure–mediator interaction—showed that problematic online gaming partially mediated the association between attention-deficit/hyperactivity and mental health issues: depressive symptoms (29.2%), anxiety (12.3%), psychotic experiences (20.6%), and diminished well-being (22.1%). The findings highlight the interplay of psychopathology, diminished inhibitory control, and addictive behaviors associated with negative consequences. Problematic online gaming may represent a modifiable mediator, warranting further intervention research to examine its potential as a treatment target.

Attention-deficit/hyperactivity disorder (ADHD) is a prevalent neurodevelopmental condition, affecting ~8% of children and adolescents globally, with boys being twice as likely to be diagnosed as girls[1]. This condition imposes a substantial economic burden worldwide through healthcare costs, educational support needs, and productivity losses[2]. Past research suggests that ADHD is associated with mental health challenges, including depression, anxiety, psychotic experiences, and diminished well-being[3–6].

Gaming disorder has been included in the International Classification of Diseases 11th Revision[7], and internet gaming disorder has been included in the Diagnostic and Statistical Manual of Mental Disorders, Fifth Edition, Text Revision (DSM-5-TR) as a condition for further study[8]. Cross-sectional studies suggested an association between problematic gaming behaviors and suicidal outcomes[9,10]. Also, a recent cohort study suggested a longitudinal association of offline video gaming and computer use (including computer-

[1]National Center of Neurology and Psychiatry, Tokyo, Japan. [2]New York University, New York, NY, USA. [3]Tokyo Metropolitan Institute of Medical Science, Tokyo, Japan. [4]King's College London, London, UK. [5]The University of Tokyo, Tokyo, Japan. [6]Kyoto University, Kyoto, Japan. ✉e-mail: zuinarita@ncnp.go.jp

based gaming) with mental health issues, although residual confounding was likely[11].

Theoretical frameworks may offer useful insights into how attention-deficit/hyperactivity and problematic online gaming could potentially contribute to negative mental health outcomes. Specifically, theoretical models introduced by Brand et al. and Dong et al.[12–14] suggest that addictive behaviors may emerge via the pathway involving psychopathology and diminished inhibitory control, potentially leading to long-term adverse consequences. Attention-deficit/hyperactivity, characterized by core features of psychopathology as well as diminished inhibitory control[8,15], aligns with the foundational elements of these models. Problematic online gaming, as a form of addictive behavior, may arise in response to external and internal triggers interacting with these factors. While such behaviors may provide short-term gratification, they may result in long-term negative outcomes, including worsened mental health. Fig. 1 illustrates the potential pathway, highlighting how psychopathology, diminished inhibitory control, and addictive behaviors contribute to negative mental health consequences in the context of attention-deficit/hyperactivity and problematic online gaming. Given the heterogeneous nature of problematic online gaming, it may, similar to other forms of addictive behaviors[8], be associated with a range of negative mental health outcomes depending on individual vulnerabilities and contextual factors.

Based on the theoretical models mentioned above, we hypothesized that attention-deficit/hyperactivity increases the likelihood of engaging in problematic online gaming; problematic online gaming contributes to mental health issues. Indeed, previous studies suggested the association between attention-deficit/hyperactivity and problematic online gaming[16–19], as well as between problematic online gaming and negative mental health outcomes[20–23]. We additionally hypothesized that problematic online gaming serves as a mediator between attention-deficit/hyperactivity and these outcomes. These hypotheses are interconnected and aim to address the following:

(1) Problematic online gaming is associated with subsequent mental health issues.

(2) Attention-deficit/hyperactivity is associated with subsequent problematic online gaming.

(3) Problematic online gaming mediates the relationship between attention-deficit/hyperactivity and mental health issues, accounting for its interaction with attention-deficit/hyperactivity.

This study explores the dynamic interplay of attention-deficit/hyperactivity and problematic online gaming associated with mental health, with a particular focus on problematic online gaming as a modifiable mediator. By examining these relationships, this study aims to provide empirical support for theoretical models and inform clinical and public health strategies. We employed causal mediation analysis to investigate the mediation effect of problematic online gaming on the relationship between attention-deficit/hyperactivity and mental health issues. This approach requires several key considerations[24]:

(a) A clear temporal order of exposure, mediator, and outcome.

(b) Control for exposure-mediator confounding and mediator-outcome confounding, as well as exposure-outcome confounding, when evaluating natural direct and indirect effects.

(c) Consideration of exposure-mediator interaction.

Unfortunately, studies employing mediation analysis often encounter these methodological challenges, which hinder their interpretations and the applicability of their findings to real-world contexts[24]. This underscores the need for more rigorous causal inference approaches to provide meaningful implications. In the present study, we aimed to address all of these considerations by using three time-point data, carefully adjusting for confounding, and incorporating the exposure-mediator interaction (the interaction between attention-deficit/hyperactivity and problematic online gaming).

## Methods
### Participants
We used data from the Tokyo Teen Cohort, a population-based cohort with multiple time points[25]. The eligibility criteria were as follows: (1) adolescents

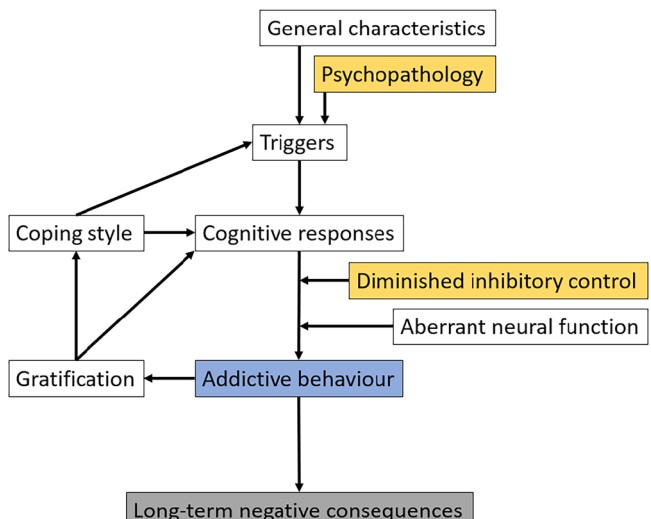

**Fig. 1 | Simplified potential pathway from attention-deficit/hyperactivity to mental health issues via problematic online gaming.** The psychopathological aspects of attention-deficit/hyperactivity may act as predisposing factors influencing triggers. These triggers can be further affected by the diminished inhibitory control associated with attention-deficit/hyperactivity, contributing to the development of addictive behaviors like problematic online gaming. Problematic online gaming, while providing immediate gratification, may lead to negative consequences.

born between September 2002 and August 2004, (2) living in one of three municipalities in Japan (Setagaya, Mitaka, or Chofu), and (3) written parental consent was obtained. Note that adolescents were included regardless of the degree of problematic online gaming. A random sampling via the basic resident register was conducted in these municipalities. A total of 3171 child-parent pairs participated at age 10 (November 2012 to January 2015), 3007 at age 12 (August 2014 to January 2017; follow-up rate: 94.8%), 2667 at age 14 (August 2014 to January 2017; follow-up rate: 84.1%), and 2616 at age 16 (February 2019 to September 2021; follow-up rate: 82.5%). Trained interviewers visited the participants' homes and provided self-report questionnaires to child-parent pairs at each time point. While we used information from ages 12, 14, and 16 in the present study, all 3171 children were included in the analysis using the imputation method, as highlighted in the "Statistical analysis" section. All procedures were approved by the Institutional Review Boards of the Tokyo Metropolitan Institute of Medical Science (12-35), the University of Tokyo (10057), and the SOKENDAI (2012002). All parents provided written informed consent. This study was not pre-registered. Data on race and ethnicity were not collected.

### Attention-deficit/hyperactivity
To assess attention-deficit/hyperactivity at age 12, we employed the Strengths and Difficulties Questionnaire hyperactivity/inattention subscale[26], a parent-report five-item measure widely used in epidemiological studies. The subscale evaluates behaviors such as restlessness, overactivity, and difficulties in sustaining attention. Each item was rated on a three-point scale: "not true," "somewhat true," and "certainly true" scored as 0, 1, and 2 points, respectively. The score was calculated by summing the responses (possible range 0–10), with higher scores indicating greater levels of attention-deficit/hyperactivity. The data demonstrated acceptable internal consistency (Cronbach's $\alpha$, 0.75).

### Problematic online gaming
We assessed problematic online gaming at age 14 using a self-reported measure employed in the previous study[27]. This measure was originally developed based on modified gambling disorder criteria, rather than the proposed internet gaming disorder criteria in the DSM-5-TR[8]. This measure included nine items with yes/no response options evaluating the presence of

behaviors or emotions associated with addiction to online games over the past 12 months. Some items, such as those related to chasing behavior and financial reliance, may reflect constructs characteristic of gambling disorder features. Details of items can be found in Supplementary Table 1. The score was calculated by summing the responses (possible range 0–9), with higher scores indicating greater levels of problematic online gaming. For dichotomization, a cut-off of four or more symptoms was used, consistent with gambling disorder criteria rather than internet gaming disorder criteria[8]. The data demonstrated limited internal consistency (Cronbach's α, 0.64), and thus the results should be interpreted with caution.

## Mental health outcomes

To assess depressive symptoms at ages 12 and 16, we employed the Short Mood and Feelings Questionnaire (SMFQ), a standard 13-item self-report measure[28,29], following the recommendation from the NICE guidance[30]. The scale evaluates depressive symptoms over the past 2 weeks. Each item had three response options: "always," "sometimes," and "never," scored with 2, 1, and 0 points, respectively. Items were summed into a composite score (possible range 0–26); a greater score suggests a higher degree of depressive symptoms. The data demonstrated strong internal consistency (Cronbach's α, 0.92).

To assess anxiety at ages 12 and 16, we utilized the Child Behavior Checklist (CBCL)[31], a widely used caregiver-reported questionnaire consisting of 118 items. To mitigate the burden on participants, we selected 84 items from the CBCL for the survey. Of these, 14 items were specifically designed to assess anxiety, derived from the CBCL anxiety scale, which initially comprised 16 items. The scale evaluates anxiety symptoms over the past 6 months. Each item was evaluated on a three-point scale: "not true," "somewhat or sometimes true," and "very true or often true." A total score was calculated by summing the responses for each item, with higher scores indicating a higher degree of anxiety (possible range 0–28). The data demonstrated acceptable internal consistency (Cronbach's α, 0.78).

We assessed psychotic experiences[32] at ages 12 and 16 using the Adolescent Psychotic-like Symptom Screener (APSS), an established seven-item self-report questionnaire[33,34]. Each item had three response options: "yes, definitely," "maybe," and "no, never," scored with 1, 0.5, and 0 points, respectively. At age 12, participants were asked about lifetime experiences, while at age 16, they were asked about experiences within the past 2 years. Items were summed into a total score (possible range 0–7); a greater score suggests a higher degree of psychotic experiences. Details of items can be found in Supplementary Table 2. The data demonstrated acceptable internal consistency (Cronbach's α, 0.71).

To assess well-being at ages 12 and 16, we employed the WHO-Five Well-Being Index (WHO-5), a standard five-item self-report questionnaire[35]. The WHO-5 evaluates well-being over the past 2 weeks, assessing positive aspects of mental health, such as mood, vitality, and general interest in life. Each item had six response options: "not at all," "a little," "somewhat," "quite a bit," "a lot," and "all of the time." A total score was calculated by summing the responses and multiplying by four (possible range 0–100), with higher scores indicating better well-being. The data demonstrated good internal consistency (Cronbach's α, 0.81).

## Covariates

We selected covariates at age 12 based on the modified disjunctive cause criterion to adjust for potential causes of the exposure, outcome, or both, excluding instrumental variables and including covariates that acted as proxies for unmeasured variables that are common causes of both the exposure and the outcome[36]. The model included age, gender, body mass index[37], IQ[38], household income[39], loneliness[40,41], physical punishment[38], relationships of mother, father, and friends[42], neighborhood cohesion[43,44], and gender nonconforming behavior[42]. Per analysis, we included the baseline level of each mental health condition, time spent gaming, and problematic internet use at age 12, thereby mitigating the possibility of reverse causation. Although direct information on problematic online gaming at age 12 was unavailable, the set of time spent gaming and problematic internet use was supposed to serve as a proxy[45] for problematic online gaming due to its overlap, and

conditioning on these variables as covariates aimed to mitigate the confounding bias caused by problematic online gaming at this age. Details on covariates are shown in Supplementary Method 1.

## Statistical analysis

All analyses were performed using R 4.4.1. First, we conducted a regression analysis to evaluate the association between problematic online gaming at age 14 and mental health at age 16, namely depressive symptoms, anxiety, psychotic experiences, and well-being. We fitted three models: Model 1 was unadjusted; Model 2 adjusted for age, gender, body mass index, IQ, and household income at age 12; and Model 3 adjusted for the variables included in Model 2, loneliness, physical punishment, relationships with mother, father, and friends, neighborhood cohesion, gender nonconformity, and the baseline level of each mental health condition at age 12.

Subsequently, we applied doubly robust estimation, dichotomizing the exposure and outcomes, while adjusting for the aforementioned covariates included in Model 3. Doubly robust estimation is a causal inference approach designed to be robust to potential model misspecification[46]. To develop a doubly robust estimator, we first fitted propensity models to calculate inverse probability weights. Using these weights, we implemented canonical link generalized linear models. While such models do not automatically yield a doubly robust estimator, we subsequently applied standardization (the g-formula) for the average causal effects (risk difference (RD) and risk ratio (RR)) to ensure the doubly robust property. Problematic online gaming was dichotomized using a cut-off score of four (low degree: 0–3; high degree: 4–9), consistent with prior research[27]. Depressive symptoms were dichotomized with a cut-off score of eight on the SMFQ[28], while anxiety was dichotomized with a T-score of 65[47]. Psychotic experiences were identified by selecting "yes, definitely" for at least one item on the APSS[34]. Diminished well-being was dichotomized using a cut-off score of 50 on the WHO-5[35].

For the doubly robust estimation, we explored effect heterogeneity across genders. To achieve this, we employed models with a product term between problematic online gaming and gender. We then computed the relative excess risk due to interaction (RERI) as an additive scale, as well as the ratio of risk ratios (RRR) as a multiplicative scale. The RERI and RRR demonstrate how the combined effects of problematic online gaming and gender influence the RR for mental health issues beyond their individual effects in both additive and multiplicative ways. The calculations for RERI and RRR are detailed in Supplementary Method 2.

To assess the robustness of the association between problematic online gaming and mental health outcomes, we conducted four sensitivity analyses. First, we performed regression analyses evaluating problematic online gaming as a dichotomous variable. Second, we conducted Poisson regression analyses, treating mental health issues as dichotomous variables. Third, for the doubly robust estimation, we analyzed E-values to estimate the degree of unmeasured confounding necessary to negate the observed associations. Fourth, we excluded loneliness from the model, considering that loneliness was assessed via the SMFQ, which was also used to examine depressive symptoms.

Next, we evaluated the association between attention-deficit/hyperactivity at age 12 and problematic online gaming at age 14. We fitted three models: Model 1 was unadjusted; Model 2 adjusted for age, gender, body mass index, IQ, and household income at age 12; and Model 3 adjusted for the variables included in Model 2, loneliness, physical punishment, relationships with mother, father, and friends, neighborhood cohesion, gender nonconformity, and the baseline level of time spent gaming and problematic internet use at age 12.

Lastly, we conducted a causal mediation analysis[48] to examine the interplay of attention-deficit/hyperactivity (exposure) and problematic online gaming (mediator) associated with subsequent mental health issues (outcomes). To ascertain the clear temporal order, we used data on attention-deficit/hyperactivity at age 12, problematic online gaming at age 14, and mental health issues at age 16. To address the exposure-mediator, mediator-outcome, and exposure-outcome confounding, we adjusted for age, gender, body mass index, IQ, household income, loneliness, physical

**Table 1 | Baseline characteristics of adolescents at age 12**

| Variables | Overall (*N* = 3171) | Participants with attention-deficit/ hyperactivity data (*N* = 2988) | Participants missing attention-deficit/ hyperactivity data (*N* = 183) |
|---|---|---|---|
| Age (years), mean (SD) | 12.2 (0.28) | 12.2 (0.28) | 12.2 (0.28) |
| Missing, *n* (%) | 4 (0.1) | 3 (0.1) | 1 (0.5) |
| Gender, *n* (%) | | | |
| Female | 1487 (46.9) | 1410 (47.2) | 77 (42.1) |
| Male | 1684 (53.1) | 1578 (52.8) | 106 (57.9) |
| Body mass index, mean (SD) | 17.9 (2.51) | 17.9 (2.50) | 18.2 (3.65) |
| Missing, *n* (%) | 617 (19.5) | 449 (15.0) | 168 (91.8) |
| IQ, mean (SD) | 110 (14.9) | 110 (14.9) | 112 (14.5) |
| Missing, *n* (%) | 631 (19.9) | 463 (15.5) | 168 (91.8) |
| Household income, JPY, *n* (%) | | | |
| < 5,000,000 | 448 (14.1) | 445 (14.9) | 3 (1.6) |
| 5,000,000–9,999,999 | 1300 (41.0) | 1293 (43.3) | 7 (3.8) |
| ≥ 10,000,000 | 681 (21.5) | 679 (22.7) | 2 (1.1) |
| Missing | 742 (23.4) | 571 (19.1) | 171 (93.4) |
| Loneliness, *n* (%) | | | |
| Never | 2121 (66.9) | 2112 (70.7) | 9 (4.9) |
| Sometimes or always | 399 (12.6) | 393 (13.2) | 6 (3.3) |
| Missing | 651 (20.5) | 483 (16.2) | 168 (91.8) |
| Physical punishment, *n* (%) | | | |
| Never or rarely | 1840 (58.0) | 1832 (61.3) | 8 (4.4) |
| Sometimes, often, or always | 906 (28.6) | 899 (30.1) | 7 (3.8) |
| Missing | 425 (13.4) | 257 (8.6) | 168 (91.8) |
| Relationship with mother, mean (SD) | 3.79 (0.986) | 3.80 (0.982) | 3.73 (1.06) |
| Missing, *n* (%) | 406 (12.8) | 392 (13.1) | 14 (7.7) |
| Relationship with father, mean (SD) | 3.43 (1.19) | 3.43 (1.19) | 3.47 (1.27) |
| Missing, *n* (%) | 406 (12.8) | 392 (13.1) | 14 (7.7) |
| Relationships with friends, mean (SD) | 3.52 (0.922) | 3.52 (0.920) | 3.47 (0.958) |
| Missing, *n* (%) | 406 (12.8) | 392 (13.1) | 14 (7.7) |
| Neighborhood cohesion, mean (SD) | 12.9 (2.89) | 12.9 (2.90) | 13.1 (2.76) |
| Missing, *n* (%) | 177 (5.6) | 170 (5.7) | 7 (3.8) |
| Gender nonconforming behavior, *n* (%) | | | |
| Not at all | 1817 (57.3) | 1706 (57.1) | 111 (60.7) |
| Somewhat, sometimes, or often | 704 (22.2) | 661 (22.1) | 43 (23.5) |
| Missing | 650 (20.5) | 621 (20.8) | 29 (15.8) |
| Time spent gaming, *n* (%) | | | |
| Less than 1 h/day | 2125 (67.0) | 2118 (70.9) | 7 (3.8) |
| 1 h/day or longer | 873 (27.5) | 864 (28.9) | 9 (4.9) |
| Missing, *n* (%) | 173 (5.5) | 6 (0.2) | 167 (91.3) |
| Problematic internet use, mean (SD) | 3.94 (4.16) | 3.92 (4.15) | 7.41 (4.81) |
| Missing, *n* (%) | 202 (6.4) | 36 (1.2) | 166 (90.7) |
| Depressive symptoms, mean (SD) | 3.84 (4.49) | 3.82 (4.47) | 6.80 (6.33) |
| Missing, *n* (%) | 692 (21.8) | 524 (17.5) | 168 (91.8) |
| Anxiety, mean (SD) | 2.39 (2.87) | 2.39 (2.87) | 1.67 (2.06) |
| Missing, *n* (%) | 446 (14.1) | 278 (9.3) | 168 (91.8) |

**Table 1 (continued) | Baseline characteristics of adolescents at age 12**

| Variables | Overall (*N* = 3171) | Participants with attention-deficit/ hyperactivity data (*N* = 2988) | Participants missing attention-deficit/ hyperactivity data (*N* = 183) |
|---|---|---|---|
| Psychotic experiences, mean (SD) | 0.581 (0.828) | 0.574 (0.819) | 0.697 (0.956) |
| Missing, *n* (%) | 632 (19.9) | 604 (20.2) | 28 (15.3) |
| Well-being, mean (SD) | 75.3 (18.9) | 75.3 (18.9) | 71.8 (18.7) |
| Missing, *n* (%) | 303 (9.6) | 139 (4.7) | 164 (89.6) |

*SD* standard deviation.

punishment, relationships with mother, father, and friends, neighborhood cohesion, gender nonconformity, as well as the baseline levels of time spent gaming, problematic internet use, and each mental health outcome condition at age 12. To capture the dynamics of mediation, we included the interaction of exposure (attention-deficit/hyperactivity) and mediator (problematic online gaming).

Addressing these key considerations, we used a two-way decomposition analysis, decomposing the total effect of attention-deficit/hyperactivity on mental health into the pure direct effect and total indirect effect[49,50].

Total Effect = Pure Direct Effect + Total Indirect Effect

- Total Effect: The total effect of a one standard deviation (SD) increase from the mean in the attention-deficit/hyperactivity score on mental health, accounting for both the pathways with and without problematic online gaming.
- Pure Direct Effect: The effect of a 1 SD increase from the mean in the attention-deficit/hyperactivity score on mental health, fixing the problematic online gaming score to the value that an individual would take if the attention-deficit/hyperactivity score were at the mean.
- Total Indirect Effect: The effect of changing the problematic online gaming score from the value that an individual would take if the attention-deficit/hyperactivity score were at the mean to the value that the same individual would take if the attention-deficit/hyperactivity score were increased by 1 SD, fixing the attention-deficit/hyperactivity score at the 1 SD increased value.

Note that we examined the effects for a 1 SD increase from the mean in attention-deficit/hyperactivity as a standard approach, while any two trajectories can be compared[51]. Proportion mediated was calculated by dividing the total indirect effect by the total effect.

Proportion Mediated = Total Indirect Effect/Total Effect

Proportion mediated represents the proportion of the total effect attributable to the causal pathway through problematic online gaming, which reflects the importance of problematic online gaming-related mechanisms in explaining the effect of attention-deficit/hyperactivity on mental health.

For valid inference of natural direct and indirect effects, four assumptions must hold[48]:

(A) No exposure-outcome confounding after the adjustments.
(B) No mediator-outcome confounding after the adjustments.
(C) No exposure-mediator confounding after the adjustments.
(D) No mediator-outcome confounders influenced by exposure.

While these assumptions are unverifiable, we sought to account for potential confounding as thoroughly as possible, as described above. As a sensitivity analysis for the mediation effect, we excluded loneliness from the model, considering that loneliness was assessed via the SMFQ, which was also used to examine depressive symptoms.

For regression analyses involving continuous outcomes, data distribution was assumed to be normal, but this was not formally tested.

**Table 2 | Association of problematic online gaming at age 14 with mental health at age 16**

| Samples and outcomes | Model 1 | | Model 2 | | Model 3 | |
|---|---|---|---|---|---|---|
| | β (per 1 SD) [95% CI] | P | β (per 1 SD) [95% CI] | P | β (per 1 SD) [95% CI] | P |
| Overall (N = 3171) | | | | | | |
| Depressive symptoms | 1.05 [0.87, 1.23] | <0.001 | 1.19 [1.01, 1.37] | <0.001 | 0.91 [0.73, 1.08] | <0.001 |
| Anxiety | 0.77 [0.65, 0.90] | <0.001 | 0.81 [0.69, 0.92] | <0.001 | 0.44 [0.34, 0.54] | <0.001 |
| Psychotic experiences | 0.09 [0.07, 0.12] | <0.001 | 0.10 [0.07, 0.12] | <0.001 | 0.09 [0.06, 0.11] | <0.001 |
| Well-being | −4.36 [−5.09, −3.62] | <0.001 | −4.52 [−5.26, −3.79] | <0.001 | −3.24 [−3.95, −2.53] | <0.001 |
| Girls (N = 1487) | | | | | | |
| Depressive symptoms | 1.45 [1.16, 1.75] | <0.001 | 1.41 [1.12, 1.71] | <0.001 | 1.08 [0.79, 1.37] | <0.001 |
| Anxiety | 0.90 [0.72, 1.09] | <0.001 | 0.88 [0.70, 1.07] | <0.001 | 0.44 [0.29, 0.59] | <0.001 |
| Psychotic experiences | 0.13 [0.09, 0.17] | <0.001 | 0.12 [0.08, 0.16] | <0.001 | 0.11 [0.07, 0.15] | <0.001 |
| Well-being | −5.33 [−6.42, −4.24] | <0.001 | −5.31 [−6.40, −4.22] | <0.001 | −3.83 [−4.89, −2.76] | <0.001 |
| Boys (N = 1684) | | | | | | |
| Depressive symptoms | 0.98 [0.78, 1.18] | <0.001 | 0.97 [0.77, 1.17] | <0.001 | 0.76 [0.56, 0.96] | <0.001 |
| Anxiety | 0.75 [0.61, 0.90] | <0.001 | 0.73 [0.59, 0.88] | <0.001 | 0.42 [0.30, 0.54] | <0.001 |
| Psychotic experiences | 0.07 [0.04, 0.10] | <0.001 | 0.07 [0.04, 0.10] | <0.001 | 0.07 [0.04, 0.10] | <0.001 |
| Well-being | −3.83 [−4.81, −2.86] | <0.001 | −3.81 [−4.79, −2.84] | <0.001 | −2.68 [−3.62, −1.74] | <0.001 |

β is reported per 1 SD increase in the exposure; its magnitude depends on each outcome's scale and is not comparable across outcomes.
Missing data were handled using random forest imputation.
Model 1 was unadjusted.
Model 2 adjusted for age, gender, body mass index, IQ, and household income at age 12.
Model 3 adjusted for the variables included in Model 2, loneliness, physical punishment, relationships with mother, father, and friends, neighborhood cohesion, gender nonconformity, attention-deficit/hyperactivity, and each mental health variable at age 12.
SD standard deviation, CI confidence interval.

Missing data were handled via random forest imputation[52]. All analyses were stratified by gender. β is reported per 1 SD increase in the exposure; its magnitude depends on each outcome's scale and is not comparable across outcomes. Confidence intervals (CIs) were calculated using the standard error for regression analysis, the percentile bootstrap approach for the doubly robust estimation, and the delta method for the mediation analysis.

## Results

### Baseline characteristics

Table 1 summarizes the baseline characteristics of the study population at age 12, comparing participants with and without attention-deficit/hyperactivity data. The mean age was 12.2 years (SD, 0.28). The sample consisted of 1487 girls (46.9%) and 1684 boys (53.1%). Participants missing attention-deficit/hyperactivity data appeared more likely to be male, had a higher degree of depressive symptoms and psychotic experiences, a lower degree of anxiety and well-being, and a higher degree of problematic internet use. Other characteristics appeared similar. The interpretation of these differences should be considered carefully, as the group missing attention-deficit/hyperactivity data had substantially higher rates of missing data across most variables (frequently exceeding 90%) compared to those with attention-deficit/hyperactivity data (typically 15–20%).

### Association between problematic online gaming and mental health

Gender-specific scores for problematic online gaming at age 14 and mental health outcomes at age 16 are shown in Supplementary Table 3. First, we conducted regression analysis to examine the association between problematic online gaming at age 14 and mental health at age 16 (Table 2 and Supplementary Fig. 1.) In Model 3 (the fully adjusted model), higher problematic online gaming scores were associated with all mental health outcomes: a higher degree of depressive symptoms (β: 0.91, 95% CI: 0.73–1.08, $p < 0.001$), a higher degree of anxiety (β: 0.44, 95% CI: 0.34–0.54, $p < 0.001$), a higher degree of psychotic experiences (β: 0.09, 95% CI: 0.06–0.11, $p < 0.001$), and a lower degree of well-being (β: −3.24, 95% CI: −3.95 to −2.53, $p < 0.001$). These associations were consistently demonstrated across genders and across Model 1 to Model 3.

Subsequently, we applied doubly robust estimation to further investigate these associations (Fig. 2 and Supplementary Table 4)[46]. A high degree of problematic online gaming was consistently associated with both increased absolute and relative risks across mental health outcomes. The fully adjusted RD and RR (95% CIs) were as follows: incident depression, 7.8% (3.0%–13.1%) and 1.62 (1.25–2.05); incident anxiety, 5.7% (2.7%–8.7%) and 1.98 (1.45–2.67); incident psychotic experiences, 5.9% (2.3%–10.8%) and 1.72 (1.30–2.47); diminished well-being, 9.6% (5.1%–14.3%) and 1.54 (1.27–1.84).

For the doubly robust estimation, we explored the interaction between problematic online gaming and gender. Both genders showed elevated risks (Fig. 2 and Supplementary Table 4). Significant additive interaction was shown for incident depression (RERI: 1.21, 95% CI: 0.11–2.55), suggesting the absolute risk increase was larger in girls (Supplementary Table 5). On the other hand, no significant additive interaction was found for other outcomes. None of the outcomes showed significant multiplicative interaction.

We conducted four sensitivity analyses to assess the robustness of the association between problematic online gaming and subsequent mental health issues. First, we performed regression analyses evaluating problematic online gaming as a dichotomous variable and found significant associations with all mental health outcomes (Supplementary Table 6). Second, we conducted Poisson regression analyses, treating mental health issues as dichotomous variables, and observed significantly increased risks for all outcomes (Supplementary Table 7). Third, we examined the E-values for the doubly robust estimation and found that substantial unmeasured confounding would be required to explain away the observed estimates (E-values for point estimates: 2.62 for incident depression, 3.37 for incident anxiety, 2.82 for incident psychotic experiences, and 2.45 for diminished well-being) (Supplementary Table 8). Fourth, we excluded loneliness from the model and found that the findings remained consistent (Supplementary Table 9). Gender-stratified sensitivity analyses showed patterns consistent with our main findings.

### Association between attention-deficit/hyperactivity and problematic online gaming

Next, we evaluated the association between attention-deficit/hyperactivity at age 12 and problematic online gaming at age 14 (Supplementary Fig. 2 and

Supplementary Table 10). Gender-specific scores for attention-deficit/hyperactivity at age 12 are shown in Supplementary Table 3. In Model 3 (the fully adjusted model), higher attention-deficit/hyperactivity scores were associated with a greater degree of problematic online gaming (adjusted $\beta$ per 1 SD: 0.18, 95% CI: 0.12–0.24, $p < 0.001$). These associations were consistently demonstrated across genders and across Model 1 to Model 3. As a sensitivity analysis, we analyzed problematic online gaming in a dichotomous manner and found that attention-deficit/hyperactivity remained significantly associated with a high degree of problematic online gaming (Supplementary Table 9).

## Causal mediation analysis

Lastly, we conducted a causal mediation analysis, addressing critical methodological considerations outlined in Table 3. Fig. 3 shows the causal diagram representing the mediation effect of problematic online gaming at age 14 on the relationship between attention-deficit/hyperactivity at age 12 and mental health outcomes at age 16. Fig. 4 and Table 4 summarize the results for mediation effects. Significant total indirect effects were observed across all mental health outcomes: depressive symptoms ($\beta$: 0.08, 95% CI: 0.04–0.11, $p < 0.001$), anxiety ($\beta$: 0.04, 95% CI: 0.02–0.06, $p < 0.001$), psychotic experiences ($\beta$: 0.01, 95% CI: 0.01–0.02, $p < 0.001$), and diminished well-being ($\beta$: 0.28, 95% CI: 0.14–0.41, $p < 0.001$). The proportion mediated was 29.2% for depressive symptoms, 12.3% for anxiety, 20.6% for psychotic experiences, and 22.1% for diminished well-being. Gender-stratified analyses showed that the mediation effects were pronounced in both girls and boys. As a sensitivity analysis, we excluded loneliness from the adjustment model and found that the results did not substantially change (Supplementary Fig. 3 and Supplementary Table 12).

## Discussion

This study provides evidence supporting our three hypotheses: (1) problematic online gaming is associated with subsequent mental health issues, (2) attention-deficit/hyperactivity is associated with subsequent problematic online gaming, and (3) problematic online gaming mediates the relationship between attention-deficit/hyperactivity and mental health issues. A high degree of problematic online gaming showed 5%–10% absolute risk

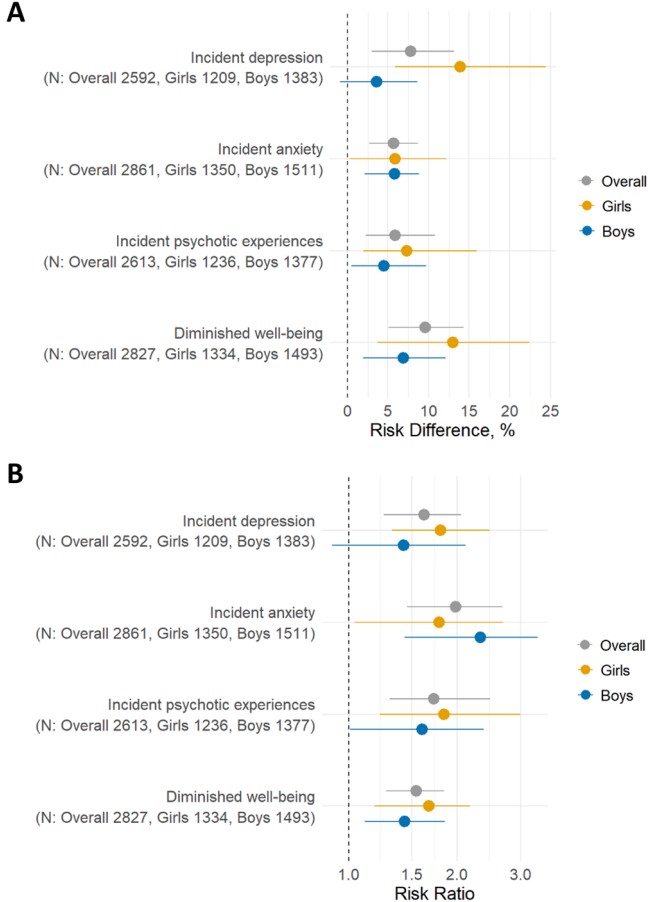

**Fig. 2 | Doubly robust estimation for the impact of problematic online gaming on mental health issues. A** Risk difference; **B** risk ratio. Mental health issues at age 16 were compared between high and low problematic online gaming groups at age 14, adjusting for a wide range of covariates (see "Methods" section). To develop a doubly robust estimator, we first fitted propensity models to calculate inverse probability weights. Using these weights, we implemented canonical link generalized linear models. While such models do not automatically yield a doubly robust estimator, we subsequently applied standardization (the g-formula) for the average causal effects to ensure the doubly robust property.

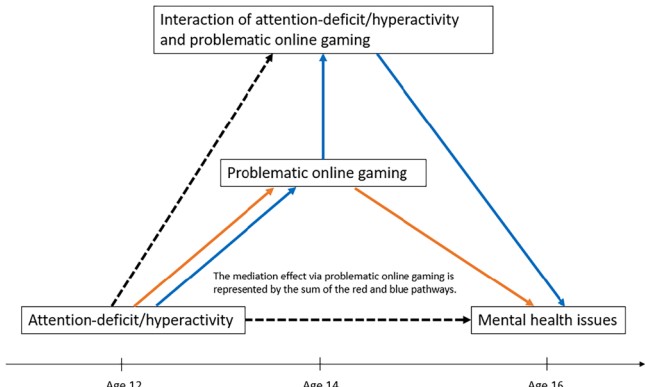

**Fig. 3 | Causal diagram representing the mediation effect of problematic online gaming at age 14 on the relationship between attention-deficit/hyperactivity at age 12 and mental health outcomes at age 16.** To capture the dynamics of mediation, we accounted for the interaction of exposure (attention-deficit/hyperactivity) and mediator (problematic online gaming).

## Table 3 | Key considerations in a causal mediation analysis involving attention-deficit/hyperactivity, problematic online gaming, and mental health

| Description | Challenges |
|---|---|
| Temporal order of exposure, mediator, and outcome | Establishing a clear temporal order of attention-deficit/hyperactivity, problematic online gaming, and mental health necessitates the inclusion of at least three time points. Failure to address this can result in reverse causation and inability to establish causal relationships. |
| Exposure-mediator confounding and mediator-outcome confounding when examining natural direct and indirect effects | Comprehensive adjustment is required for attention-deficit/hyperactivity-problematic online gaming, problematic online gaming-mental health, and attention-deficit/hyperactivity-mental health confounding. Without this, confounding bias and reverse causation can compromise the findings. |
| Exposure-mediator interaction | The interaction between attention-deficit/hyperactivity and problematic online gaming likely exists in real-world scenarios and must be accounted for. Ignoring this can lead to model misspecification and failure to capture the dynamics of mediation. |

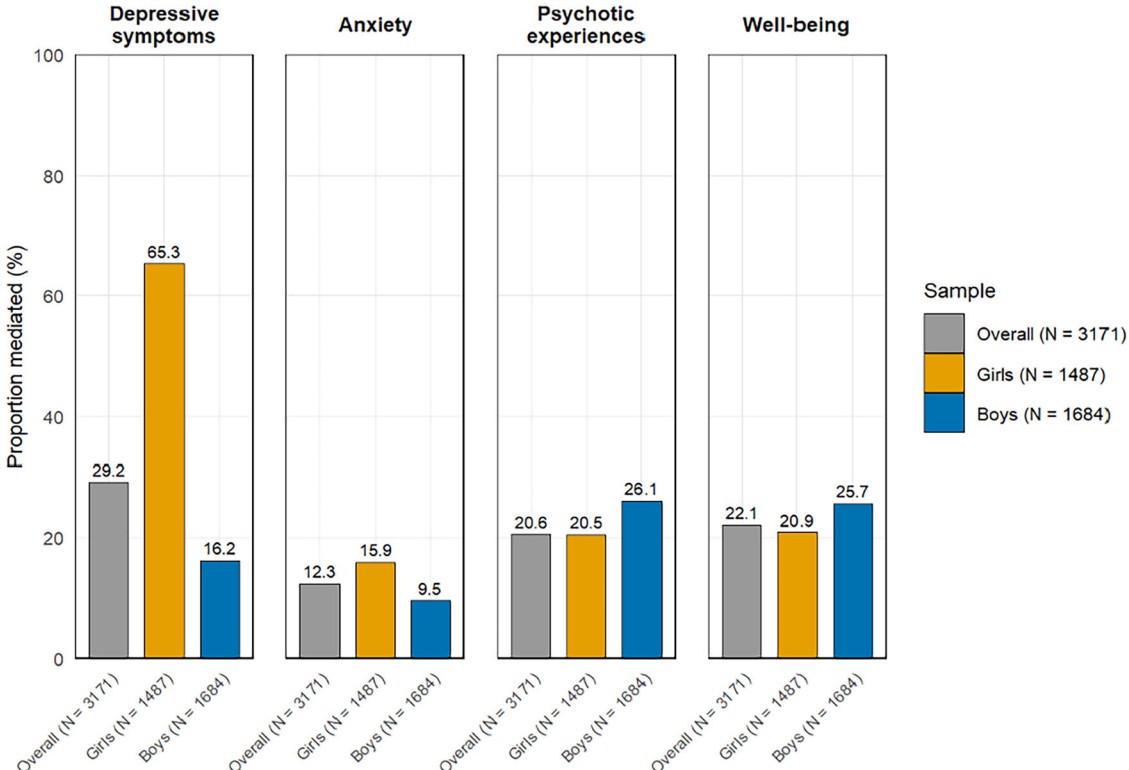

**Fig. 4 | Proportion mediated by problematic online gaming in the relationship between attention-deficit/hyperactivity and mental health issues.** Bars represent the proportion of the total effect of attention-deficit/hyperactivity at age 12 on mental health issues at age 16 that is attributable to the causal pathway through problematic online gaming at age 14. This reflects the importance of problematic online gaming–related mechanisms in explaining the effect of attention-deficit/hyperactivity on mental health issues.

**Table 4 | Mediation effect of problematic online gaming at age 14 on the relationships between attention-deficit and hyperactivity at age 12 and mental health at age 16**

| Samples and outcomes | Total effect β (per 1 SD) [95% CI] | Pure direct effect β (per 1 SD) [95% CI] | Total indirect effect β (per 1 SD) [95% CI] | P for total indirect effect |
|---|---|---|---|---|
| Overall (N = 3171) | | | | |
| Depressive symptoms | 0.26 [0.07, 0.45] | 0.18 [0.00, 0.37] | 0.08 [0.04, 0.11] | <0.001 |
| Anxiety | 0.32 [0.22, 0.43] | 0.28 [0.18, 0.39] | 0.04 [0.02, 0.06] | <0.001 |
| Psychotic experiences | 0.06 [0.03, 0.08] | 0.05 [0.02, 0.07] | 0.01 [0.01, 0.02] | <0.001 |
| Well-being (reverse scored) | 1.26 [0.50, 2.03] | 0.98 [0.22, 1.75] | 0.28 [0.14, 0.41] | <0.001 |
| Girls (N = 1487) | | | | |
| Depressive symptoms | 0.16 [−0.15, 0.47] | 0.05 [−0.25, 0.37] | 0.10 [0.04, 0.17] | 0.001 |
| Anxiety | 0.30 [0.13, 0.46] | 0.25 [0.08, 0.41] | 0.05 [0.02, 0.08] | 0.003 |
| Psychotic experiences | 0.08 [0.03, 0.12] | 0.06 [0.02, 0.10] | 0.02 [0.01, 0.03] | <0.001 |
| Well-being (reverse scored) | 1.66 [0.53, 2.79] | 1.31 [0.18, 2.45] | 0.35 [0.13, 0.56] | 0.002 |
| Boys (N = 1684) | | | | |
| Depressive symptoms | 0.38 [0.16, 0.59] | 0.32 [0.10, 0.53] | 0.06 [0.02, 0.11] | 0.006 |
| Anxiety | 0.36 [0.22, 0.49] | 0.32 [0.19, 0.45] | 0.03 [0.01, 0.06] | 0.006 |
| Psychotic experiences | 0.03 [0.00, 0.07] | 0.02 [−0.01, 0.06] | 0.01 [0.003, 0.02] | 0.003 |
| Well-being (reverse scored) | 0.94 [−0.07, 1.94] | 0.70 [−0.31, 1.70] | 0.24 [0.07, 0.41] | 0.005 |

See the "Methods" section for a detailed interpretation of β.

The analysis accounted for the interaction of attention-deficit/hyperactivity and problematic online gaming.

Missing data were handled using random forest imputation.

The model adjusted for age, gender, body mass index, IQ, household income, loneliness, physical punishment, relationships with mother, father, and friends, neighborhood cohesion, gender nonconformity, time spent gaming, problematic internet use, and each mental health variable at age 12.

CIs were computed using the delta method.

*SD* standard deviation, *CI* confidence interval.

increases and 1.5–2-fold relative risk increases. These findings are consistent with previous studies showing that problematic online gaming was associated with later mental health issues [20–23]. The results were robust in multiple sensitivity analyses. Although we found no significant multiplicative interaction, possibly due to limited statistical power, the absolute risk increase in depression was larger in girls. These findings align with our previous study [53], which showed a greater impact of problematic internet use on depression in girls. While the reason for the gender-specific results on depression is beyond the scope of this paper, Japanese men are reportedly more likely than women to own and play gaming consoles and tend to spend more time playing video games [54]. Some platforms are more commonly linked to solitary play, whereas others are often used in in-person social contexts [54]. These differences might be related to heterogeneous patterns of online gaming across genders.

Our findings indicated that attention-deficit/hyperactivity was significantly associated with subsequent problematic online gaming, consistent with previous studies [16–19]. Problematic online gaming, in turn, mediated the relationship between attention-deficit/hyperactivity and adverse mental health outcomes. In accordance with the theoretical model [12,13], the psychopathological aspects of attention-deficit/hyperactivity may act as predisposing factors influencing triggers. These triggers can be further affected by the diminished inhibitory control associated with attention-deficit/hyperactivity, contributing to the development of addictive behaviors like problematic online gaming. Problematic online gaming, while providing immediate gratification, may lead to negative consequences. To address key methodological considerations, we utilized three time-point data at ages 12, 14, and 16, ensuring a clear temporal order among exposure, mediator, and outcome, which allowed us to establish the directionality of these relationships. We also accounted for confounding factors as comprehensively as possible to enhance the robustness of our findings. Furthermore, our model addressed the interaction between attention-deficit/hyperactivity and problematic online gaming, thereby capturing the dynamics of mediation that are likely to exist in real-world settings. By addressing these key considerations, our findings provide robust empirical support for the existing theoretical models, emphasizing the complex role of psychopathology, diminished inhibitory control, and addictive behavior in leading to long-term negative outcomes. Pharmacological treatment for attention-deficit/hyperactivity is generally effective and widely used [55], but it does not necessarily normalize symptoms [56], i.e., the symptoms could remain only partially modifiable. In such cases, problematic online gaming may represent a modifiable mediator [57]. Further intervention study is warranted to examine its potential as a treatment target.

This study builds on and complements prior research emphasizing the positive aspects of gaming, such as fostering social connections and emotional regulation. A recent study employing a robust causal inference approach demonstrated a positive effect of gaming on mental health [54]. On the other hand, this study also highlighted the heterogeneity of these effects; for instance, the benefits diminished with longer gaming durations, and some early teens using platforms like the PlayStation 5 did not experience positive impacts. Thus, we believe our findings complement the authors' study by showing that there are populations experiencing the negative impacts of gaming, depending on how they engage with it.

## Limitations

This study has several limitations. First, despite the longitudinal design and the rich confounder adjustment, the possibility of residual confounding remains. Valid inference in causal mediation analysis relies on no-confounding assumptions (see the "Methods" section), but these assumptions are unverifiable. Specifically, we used the set of time spent gaming and problematic internet use as a proxy for problematic online gaming at age 12; however, it is unverifiable whether adjusting for this set was sufficient to account for the baseline level of problematic online gaming. Second, the potential for measurement bias exists, particularly with self-reported measures, which might not accurately capture the actual pathology as a clinical interview would. Also, the standardized measures used in this study were

not formally validated in Japanese, which could further contribute to measurement bias. Moreover, the problematic online gaming measure demonstrated limited internal consistency, which warrants caution when interpreting the findings. Third, we lacked data on potentially important factors, e.g., substance use [58] and physical injuries [59,60]. Also, we evaluated physical punishment as a proxy for adverse childhood experiences [61,62] but did not consider other stressful or traumatic exposures, such as emotional neglect. Fourth, the transportability of our findings may be limited; the sample, predominantly of Asian ethnicity from the highly urbanized city, may not represent rural or diverse racial demographics well. Lastly, the effect of problematic online gaming could be better corroborated using the framework of target trial emulation. This framework can mitigate the influence of biases such as immortal time bias caused by the misalignment of the start of follow-up, eligibility, and treatment assignment (in this case, the initiation of problematic online gaming) [63]. For example, future studies should focus on the initiation of problematic online gaming rather than problematic online gaming at a particular time point, in the same way as randomized controlled trials examine the initiation of treatment. Achieving this would require consistent and frequent recording of measurements over time.

This study advances our understanding of the interplay of attention-deficit/hyperactivity and problematic online gaming associated with negative mental health outcomes, supporting existing theoretical models. Clinicians and educators should focus on identifying and addressing early signs of problematic online gaming. Problematic online gaming may represent a modifiable mediator, warranting further intervention research to examine its potential as a treatment target.

## Data availability

The dataset containing the numerical estimates used to generate Fig. 2 (risk differences and risk ratios with 95% confidence intervals) is available on Zenodo: https://doi.org/10.5281/zenodo.15760926 [64]. Data described in the manuscript will be made available upon request pending application and approval.

## Code availability

The code used in this study is archived on Zenodo: https://doi.org/10.5281/zenodo.15760841 [65]. This version corresponds to the GitHub repository at https://github.com/ZuiCNarita/DR-and-CMA250122 and reflects the analyses reported in the manuscript.

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

## Acknowledgements

This project was funded by Japan Society for the Promotion of Science (JP20H01777, JP20H03951, JP21H05171, JP21H05173, JP21H05174, JP21K10487, JP22H05211, JP23H03174, JP23H05472, JP24H00666, JP24H00917, JP25K20564, and JP24K16821), UTokyo Institute for Diversity and Adaptation of Human Mind (UTIDAHM), the International Research Center for Neurointelligence (WPI-IRCN) at the University of Tokyo Institutes for Advanced Study (UTIAS), the Japan Science and Technology Agency (JPMJPF2105), and the Ministry of Health Labour and Welfare (23AB1002). The funders had no role in study design, data collection and analysis, decision to publish, or preparation of the manuscript.

## Author contributions

Zui C. Narita analyzed the data and wrote the initial draft of the manuscript. Syudo Yamasaki, Mitsuhiro Miyashita, and Satoshi Yamaguchi contributed to data curation. Shuntaro Ando, Jordan DeVylder, Toshiaki A. Furukawa, Kiyoto Kasai, and Atsushi Nishida provided supervision and conceptual input. Gemma Knowles and Daniel Stanyon contributed to the interpretation of findings. All authors reviewed, revised, and approved the final version of the manuscript.

## Competing interests

T.A.F. reports personal fees from Boehringer-Ingelheim, Daiichi Sankyo, DT Axis, Micron, Shionogi, SONY, and UpToDate, and a grant from DT Axis and Shionogi, outside the submitted work. In addition, T.A.F. has a patent 7448125 and a pending patent 2022-082495, and has licensed intellectual properties for Kokoro-app to Mitsubishi-Tanabe. K.K. reports grants from Teijin, Takeda, Lily, Otsuka, Daiichi Sankyo, Shionogi, Tanabe-Mitsubishi, and Sumitomo; and personal fees from Sumitomo, Meiji-Seika, Ricoh, Fuji-film Wako, Takeda, Otsuka, and Astellas outside the submitted work. The other authors declared no conflicts of interest.

## Ethical approval

All procedures were approved by the Institutional Review Boards of the Tokyo Metropolitan Institute of Medical Science (12-35), the University of Tokyo (10057), and the SOKENDAI (2012002). All parents provided written informed consent.
