## [Transparent Peer Review file · Communications Psychology]

Problematic online gaming mediates the association between attention-deficit/hyperactivity and subsequent mental health issues in adolescents

Corresponding Author: Dr Zui Narita

Version 0:

Decision Letter:

Dear Dr Narita,

Thank you for your patience during the peer-review process. Your manuscript titled "Interplay of attention-deficit/hyperactivity and problematic online gaming on subsequent mental health issues in adolescents" has now been seen by 3 reviewers, and I include their comments at the end of this message. They find your work of interest but raised some important points. We are interested in the possibility of publishing your study in *Communications Psychology*, but would like to consider your responses to these concerns and assess a revised manuscript before we make a final decision on publication.

We therefore invite you to revise and resubmit your manuscript, along with a point-by-point response to the reviewers. Please highlight all changes in the manuscript text file.

Editorially, we consider it crucial that the reviewers' concerns regarding the validity of the problematic online gaming measures are thoroughly addressed in the revised manuscript. In addition, the advance of the current study in comparison to existing longitudinal studies on ADHD and problematic online gaming should be clearly demonstrated in the revised manuscript.

I am attaching an Editorial Requests Table that details critical reporting requirements for the revised manuscript. Please attend to each item and ensure your manuscript is fully compliant. If your revised manuscript is not aligned with these requests on major issues, such as those concerning statistics, it may be returned to you for further revisions without re-review.

Please submit the following items:

- Revised manuscript
- Point-by-point response to the referees' comments
- Cover letter (as a separate document)
- <https://www.nature.com/documents/nr-reporting-summary.zip>>Nature Research Reporting Summary
- <https://www.nature.com/documents/nr-editorial-policy-checklist.pdf>>Editorial Policy Checklist
- Completed Editorial Request Table (attached).

via this link: Link Redacted .

Additional guidance is available in our style and formatting guide Communications Psychology formatting guide.

Best regards,

Troy Lui

Troy Lui, PhD
Associate Editor
Communications Psychology

REVIEWER EXPERTISE:

Reviewer #1: ADHD, Online gaming, Mediation analysis

Reviewer #2: neurodevelopmental disorders, gaming

Reviewer #3: adolescent, ADHD, internet gaming, mediation

REVIEWER REPORTS:

Reviewer #1 (Remarks to the Author):

Dear Zui C Narita (and co-authors),

I found your paper titled as "Interplay of attention-deficit/hyperactivity and problematic online gaming on subsequent mental health issues in adolescents" very fascinating and timely. This paper aimed at understanding the potential causes behind the mental health deterioration of adolescents by demonstrating the process of how the presence of ADHD symptoms are leading to common negative psychological outcomes. Regarding the high quality of this paper not only its relevance, but additionally its high methodological rigor is also noteworthy. In the research paper, the authors raise several common methodological issues of previous studies which limit the possibility of establishing more reliable causal inferences (e.g. inappropriate designs or lack of proper handling of covariates), all which they successfully manage. The paper itself is clearly written and includes a well detailed and transparent description of their methodology, results, limitations, practical implications and even directing further research. Besides supporting the publication of the paper, I do have some suggestions which could potentially improve the quality, mostly (but not only) concerning its theoretical underpinnings and implications.

1. Introduction

The author's statement: "Attention deficit/hyperactivity is often refractory to pharmacotherapy and thus unmodifiable or only partially modifiable."

Comment: The authors provide only one supporting evidence for this general statement. The supporting evidence strictly focuses on aggressive behavior, which does not cover the care needs of ADHD affected youth and is not identical to ADHD symptoms. I recommend mentioning studies where the core symptoms are targeted (several systematic reviews and meta-analyses are already published in this topic). This sentence also gives a bit of a misleading impression that no effective alternatives (such as computerized training) are available currently.

The author's statement: "In contrast, POG is a modifiable mediator, and demonstrating its mediation effect could point to potential interventions to mitigate the impact of attention deficit/hyperactivity on mental health issues."

Comment: The authors do not provide any supporting evidence for POG being modifiable. Furthermore, the authors also do not provide evidence for POG being more effectively treated compared to ADHD.

The author's statement: "The choice of psychotic experiences, along with depressive symptoms, anxiety, and well-being, was motivated by the fact that while some individuals may hold prejudice against psychotic it is precisely for this reason that we impartially evaluated this common condition during adolescence."

Comment: I do not understand this argument. The inclusion of any expected outcome should be based at least on previous comorbidity studies. Additionally, the paper would greatly benefit from describing the theorized process of how POG could result in different negative mental health outcomes (either in the introduction or in the discussion).

General comment for the introduction: The introduction is missing almost (if not all) longitudinal studies focused on the prospective relationship between ADHD and POG or POG and its negative consequences on mental health. As this study is partially aimed at determining a temporal order among these variables, what is already known about the direction of effect must be presented in the introduction.

2. Results

The author's statement: "At age 14, boys had a higher mean POG score compared to girls (1.98 vs. 1.52), while at age 16, boys had lower mean scores for depressive symptoms (2.72 vs. 4.85), anxiety (2.26 vs. 2.96), and psychotic experiences (0.24 vs. 0.27), as well as a higher mean score for well-being (67.4 vs. 65.5)."

"At age 12, boys had a higher mean attention-deficit/hyperactivity score compared to girls (3.12 vs. 2.26)."

Comment: When means are reported in your study, please also report standard deviations. Also please be cautious with statements regarding group differences without inferential statistics. Despite most of the differences described are seemingly evident based on the apparent mean differences, others should be referred to as being "close to equal" or "similar".

The author's statement: "Figure 4 and Table 4 summarize the results for mediation effects. Significant total indirect effects and reasonable proportion mediated were observed across all mental health outcomes (depressive symptoms: 29.2%; anxiety: 12.3%; psychotic experiences: 20.6%; diminished well-being: 22.1%)."

Comment: Meanwhile reporting the proportions mediated is valuable, I would recommend at least briefly reflecting on the differences in predicting your outcomes. The models were clearly more effective in the prediction of well-being of the respondents, compared to having psychotic experiences.

3. Discussion

The author's statement: "While the reason for the gender-specific results on depression is beyond the scope of this paper, the types of online activities and patterns of engagement may differ between girls and boys."

Comment: I recommend briefly discussing these findings. It would be beneficial to see if this effect is expected to be generalizable to other cultures, or it may be dependent only on Japanese population. Answering some questions would help others understand if these findings could be population-specific:

- Are females similarly engaged in gaming in Japan as males?

- What is the form of gaming in Japan (e.g. are people tend to play alone on personal consoles/computers potentially leading to isolation from non-virtual social experiences or is it more socially embedded, such as in case of South-Korean PC bangs)?

- Are there gender related differences in game platform or genre preference in Japan?

General comment for the discussion: The discussion is mainly repetition of the results and methods section. Results should only be repeated to clarify if specific results are discussed. Repeating that "...attention-deficit/hyperactivity is often refractory to pharmacotherapy and thus not always modifiable, POG represents a modifiable mediator, highlighting potential intervention points to mitigate its impact on mental health." is redundant. I would recommend only mentioning this in the discussion, as practical implications are generally expected to be in this section. The main function of the discussion section is the theoretical explanation of results, comparison with other similar findings and reflection on hypotheses and contradictions.

4. Methods

The author's statement: Attention-deficit/hyperactivity measurement

Comment: No reliability reported for the subscale used. Was this data self-reported, parent reported or did multiple informants provide information?

The author's statement: Problematic online gaming measurement

Comment: The authors wrote "We assessed POG at age 14 using a self-reported measure employed in the previous study." which previous study was not introduced earlier. It is not good practice for referencing measures as it makes it difficult for the potential reader to identify the scale. Authors could simply say "DSM criteria was used" and reference the DSM version where it is originated from. It must be mentioned that it is not the Internet gaming disorder (IGD) criteria, but a modified gambling disorder criteria was used to measure disordered gaming. This measurement has limitations, such as it measures some symptoms which were not incorporated or heavily changed as part of IGD criteria:

- "When not getting what they want in an online game (e.g., they did not get the item they want or did not clear the event), often returning on another day to accomplish this."

This symptom is seemingly a mixture of "chasing losses" criteria of gambling and the continuation of gaming IGD criteria (but it missed an important aspect, that continuation occurs despite the user experiences negative consequences).

- "Relies on others to provide relief from desperate situations caused by online gaming (e.g., borrowing money)."

Financial problems are evidently more prevalent among problem gamblers (while the overlap between gaming and gambling features is increasingly more visible).

An indication of non-relevance of these symptoms is reflected as both criteria are rarely reported among gamers compared to others (Ide et al., 2021).

This problem is not major and due to IGD criteria first being introduced to DSM in 2013 (APA, 2013), inclusion of this criteria in the current study would have been complicated (considering the necessity of translation, psychometric validation and the submission of a modified ethical permission for the study). But it has to be also mentioned that for dichotomization, gambling criteria cut-off was used (presence of 4 symptoms) instead of IGD criteria cut-off (presence of 5 symptoms). These limitations must be mentioned.

References cited:

Ide, S. et al. Adolescent Problem Gaming and Loot Box Purchasing in Video Games: Cross-sectional Observational Study

Using Population-Based Cohort Data. JMIR Serious Games 9, e23886 (2021).

American Psychiatric Association. (2013). Diagnostic and statistical manual of mental disorders: DSM-5. American psychiatric association.

The author's statement: Mental health outcomes measurement

Comment: What was the timeframe of measurement (past week/month/year)? This is a question for both depression and anxiety measurement. Additionally, for well-being measurement, the 2-week timeframe of measurement is a bit unjustified for a longitudinal study of this length.

5. Supplementary materials

The author's statement: Supplementary Method 1 | Details on the measurement of covariates.

Comment: the authors dichotomized time spent gaming using the following rule: "less than one hour/day vs. one hour/day or longer" to create a proxy for POG which can be entered as covariate at time 1. This criterion is not justified by earlier research, as an average daily 1-hour videogame use is associated with 0 symptoms of disordered gaming (Pontes et al., 2022, Katz et al., 2024).

Furthermore, while it is understandable that the detailed description of covariate measurement would further lengthen this already long document, I would appreciate if the reliability of the covariate measurement tools would be reported.

References cited:

Pontes, H. M., Schivinski, B., Kannen, C., & Montag, C. (2022). The interplay between time spent gaming and disordered gaming: A large-scale world-wide study. *Social Science & Medicine*, 296, 114721.

Katz, D., Horváth, Z., Pontes, H. M., Koncz, P., Demetrovics, Z., & Király, O. (2024). How much gaming is too much? An analysis based on psychological distress. *Journal of Behavioral Addictions*, 13(3), 716-728.

The author's statement: Supplementary Fig. 1 | Association between problematic online gaming at age 14 and mental health at age 16.

Comment: High resolution versions must be inserted, because the figure texts in the current version cannot be read.

The author's statement: Supplementary Table 4 | Sensitivity analysis for the association between problematic online gaming at age 14 and mental health at age 16: analyzing mental health as dichotomous variables.

Comment: naming dichotomous variables in a way that reflects their categorical nature would be better (e.g. instead of "depressive symptoms" using "probable depression", instead of "anxiety" using "high anxiety").

General comment for the whole paper: Disordered gaming is referred to as "POG engagement", which is a strange choice of wording. I would recommend referring to it as "severity", as a clinical disorder was being measured, not the intensity or frequency of participation in a recreational activity (as it would be if time spent playing games would have been measured, as an outcome).

I wish you good luck with publishing your results.

Patrik Koncz

Research Group Member, Lecturer

Addiction Research Group

Eötvös Loránd University, Faculty of Education and Psychology

Reviewer #2 (Remarks to the Author):

Thank you for the opportunity to review the submission of the journal article "Interplay of attention-deficit/hyperactivity and problematic online gaming on subsequent mental health issues in adolescents" (COMMSPSYCHOL-25-0086-T) in *Communications Psychology*.

This article poses an interesting and clinically relevant questions about the relationship between ADHD and mental health concerns and its mediation by problematic online gaming. The article is well aligned with the journal's scope of publishing brief yet impactful research studies and could be published after minor revisions.

The authors used existing data to conduct a robust mediation analysis and took additional steps to ensure the robustness of the analysis. The outcomes of their analysis are clear, and likely to be applicable to similar populations, making it worthwhile to share this analysis with the readers of *Communications Psychology*. Recommendations to improve the article were for the most part fairly conceptual, and the authors may choose to enact these as they see fit.

I divide my reviews into key points and minor comments, as most of the minor comments are issues that can be fixed relatively quickly, with major points often requiring further thought.

Key Points:

1. Overstatement of treatment-resistant ADHD: Pharmacological treatment of ADHD is actually one of the most effective psychiatric pharmacological interventions available, so while it is true that some cases are treatment-refractory I think stating that this happens "often" is over-stating the claim (Mechler et al., 2022). I don't think the authors need to claim that ADHD is hard to modify for them to argue that problematic gaming is an important target, as problematic gaming is itself associated with negative outcomes (as the authors highlight). Similarly, psychological therapies which are often helpful adjuncts in the treatment of ADHD are not targeting "core symptoms" of ADHD, but rather the management of the disorder – and I see a similar role for interventions targeting problematic gaming.

2. Sample representativeness: It would be good to understand who the sample is meant to represent. For example, was the

final sample from the three municipalities selected broadly similar to the population of adolescents in the Greater Tokyo area and therefore likely to represent urban middle-class Japanese adolescents with mild-to-moderate ADHD symptoms (given the high rate of drop-out from those with high ADHD symptoms)?

3. Problematic online gaming measure: Given this is a custom measure further information is warranted before treating the summative score as an adequate summation of the items in the scale. For example, the authors do not report whether the questions load onto a single factor (in either this study or the study cited as the source of the measure – reference 31); likewise, it may be helpful to include a standardised Cronbach alpha value as calculated on the basis of the participants' responses to the POG measure, as was reported for the mental health outcomes.

4. Considerations regarding gender differences: The authors may want to consider downplaying gender differences given findings were highly significant across genders, and there are important reporting differences between girls and boys (with girls more likely to report mood symptoms in the first place). At the same time, there is some evidence suggesting greater "addictiveness" in girls when engaging with specific forms of substance use (and possibly other behavioural addictions). In my view this would amount to more tentative language regarding the gender-based differences.

Minor Comments:

1. There's a period missing at the end of aim 3.
2. Page 5, Line 80: The sentence regarding the "choice of psychotic experiences" seems out of place, as by this point the reader is not aware of what is or is not included in the assessment. Please consider rewording this statement.
3. The robustness of the statistical analysis is encouraging to see.
4. Ensure the accessibility of Figures by testing whether chosen colour schemes are able to be accurately perceived by individuals with colour impairments.

Reviewer #3 (Remarks to the Author):

The study entitled "Interplay of attention-deficit/hyperactivity and problematic online gaming on subsequent mental health issues in adolescents" had some strengths, including a representative and relatively large sample (n=3171), a longitudinal design with four timepoints, several different standardized instruments assessing mental health (including SMFQ, CBCL, APSS, and WHO-5), and rigorous data analyses. The authors found that problematic online gaming is a significant mediator in the longitudinal relationships between ADHD and mental health issues. Although I appreciate the authors' findings and above-mentioned strengths, I have some concerns in the present study. Below please see my specific comments.

1. The authors claimed that "However, evidence specifically examining the impact of problematic online gaming (POG) on mental health issues remains limited"; however, this is inaccurate. There is ample evidence showing the relationship between problematic online gaming and mental health issues, including population with ADHD and some longitudinal studies on other populations. Below are some references I know (only on ADHD and longitudinal studies); however, I believe that the authors could find a lot in the databases, especially using cross-sectional design.

On ADHD:

Chen, C.-Y., Lee, K.-Y., Fung, X. C. C., Chen, J.-K., Lai, Y.-C., Potenza, M. N., Chang, K.-C., Fang C.-Y., Pakpour, A. H., & Lin, C.-Y. (2024). Problematic use of internet associates with poor quality of life via psychological distress in individuals with ADHD. *Psychology Research and Behavior Management*, 17, 443-455.

Lee, K.-Y., Chen, C.-Y., Chen, J.-K., Liu, C.-C., Chang, K.-C., Fung, X. C. C., Chen, J.-S., Kao, Y.-C., Potenza, M. N., Pakpour, A. H., Lin, C.-Y. (2023). Exploring mediational roles for self-stigma in associations between types of problematic use of internet and psychological distress in youth with ADHD. *Research in Developmental Disabilities*, 133, 104410.

Using longitudinal design:

Chang, C.-W., Huang, R.-Y., Strong, C., Lin, Y.-C., Tsai, M.-C., Chen, I.-H., Lin, C.-Y., Pakpour, A. H., & Griffiths, M. D. (2022). Reciprocal relationships between problematic social media use, problematic gaming, and psychological distress among university students: A nine-month longitudinal study. *Frontiers in Public Health*, 10, 858482.

Chen, I.-H., Lin, Y.-C., Lin, C.-Y., Wang, W.-C., & Gamble, J. H. (2022). The trajectory of psychological distress and problematic Internet gaming among primary school boys: a longitudinal study across different periods of COVID-19 in China. *Journal of Men's Health*, 18(3), 070.

Chen, C.-Y., Chen, I.-H., Hou, W.-L., Potenza, M. N., O'Brien, K. S., Lin, C.-Y., & Latner, J. D. (2022). The relationship between children's problematic Internet-related behaviors and psychological distress during the onset of the COVID-19 pandemic: A longitudinal study. *Journal of Addiction Medicine*, 16, e73-e80.

Chen, I.-H., Chen, C.-Y., Pakpour, A. H., Griffiths, M. D., Lin, C.-Y., Li, X.-D., Tsang, H. W. H. (2021). Problematic internet-related behaviors mediate the associations between levels of internet engagement and distress among schoolchildren during COVID-19 lockdown: A longitudinal structural equation modeling study. *Journal of Behavioral Addictions*, 10(1), 135-148.

2. I disagree with the statement "For POG, we used symptom-based measures to differentiate it from online gaming in general, thereby addressing the limitations of past research". Many studies have used DSM-5-based measure (e.g., Internet Gaming Disorder Scale) to examine the relationship between problematic gaming and mental health issues.

3. Lines 89-92, please use citation to support the key considerations.

4. The statement "Unfortunately, even in leading behavioral science journals, studies employing mediation analysis often encounter these methodological challenges" is vague as (i) it is unclear what "leading behavioral science journals" are and (ii) there are no citations to support this statement.

5. The authors used the abbreviation of PIU but did not define it as problematic internet use for its first-time use.

6. Following the previous comment, it is unclear how the authors collected PIU (i.e., what measure was used).
7. Results section repeats some information from Methods.
8. I suppose that the authors used the 9 criteria defined by the DSM-5 to assess POG. However, this is not clearly stated.
9. It is unclear where the participants complete the questionnaires, and how the procedure was done.
10. It is unclear if all the standardized measures have been translated into Japanese with validity evidence.
11. It is unclear how the authors assessed some covariates (e.g., IQ, loneliness, physical punishment neighborhood cohesion, and gender nonconforming behavior).
12. I would suggest using years instead of months to present age. If they are toddlers or young children, using months is better. However, they are already over 10 years and interpreting age using months for this population is non-intuitive.
13. I would suggest deleting some overlapped description in Results section. That is, if the information is not that important and can be read in Tables, the authors need not to mention the values/statistics in Results section.
14. The inclusion of sensitivity analyses is a strength.
15. I believe that the authors have to search the literature again to strengthen their Discussion section (i.e., using the literature on the relationship between problematic gaming and mental health issues to communicate with the present findings).

Version 1:

Decision Letter:

Dear Dr Narita,

Your manuscript titled "Problematic online gaming mediates the association between attention-deficit/hyperactivity and subsequent mental health issues in adolescents" has now been seen by our reviewers, whose comments appear below. In light of their advice I am delighted to say that we are happy, in principle, to publish a suitably revised version in Communications Psychology.

We therefore invite you to revise your paper one last time to address the remaining concerns of our reviewers and a list of editorial requests. At the same time we ask that you edit your manuscript to comply with our format requirements and to maximise the accessibility and therefore the impact of your work.

EDITORIAL REQUESTS:

SUBMISSION INFORMATION:

OPEN ACCESS:

* DATA AVAILABILITY:

Link Redacted

Best regards,

Troy Lui

Troy Lui, PhD
Associate Editor
Communications Psychology

REVIEWERS' COMMENTS:

Reviewer #1 (Remarks to the Author):

Dear Zui C Narita (and co-authors),

Your paper has developed significantly thanks to the revision. I do appreciate your transparency regarding the validity of POG measurement, I think in the revised version the manuscript fairly reflects on this limitation. Furthermore, I still have one minor concern regarding one of the applied solutions, but besides that, I recommend your paper for publication. In my response I copied the earlier discussion first (your "Earlier author's statement", my "Earlier reviewer comment"), then your response to the comments and my new comments at the end.

Earlier author's statement: "In contrast, POG is a modifiable mediator, and demonstrating its mediation effect could point to potential interventions to mitigate the impact of attention deficit/hyperactivity on mental health issues."

Earlier reviewer comment: The authors do not provide any supporting evidence for POG being modifiable. Furthermore, the authors also do not provide evidence for POG being more effectively treated compared to ADHD.

Response: We agree with the reviewer's concern. As highlighted in the previous response, we have removed these sentences.

New comment: I do think that POG is modifiable (at least on short-term) (see Stevens et al., 2019), the problem was not supporting it by a proper study.

Reference cited: Stevens, M. W., King, D. L., Dorstyn, D., & Delfabbro, P. H. (2019). Cognitive-behavioral therapy for Internet gaming disorder: A systematic review and meta-analysis. *Clinical psychology & psychotherapy*, 26(2), 191-203.

Wishing you good luck with the publication of your results,
Patrik Koncz
Research Group Member, Lecturer
Addiction Research Group
Eötvös Loránd University, Faculty of Education and Psychology

Reviewer #2 (Remarks to the Author):

Thank you for the chance to review the authors' responses to my comments.

I am satisfied that my comments have been addressed carefully and successfully.

Reviewer #3 (Remarks to the Author):

The authors have satisfactorily addressed my previous comments. I am happy with the revision and the presentation now looks good to me.

Editor:

Thank you for your patience during the peer-review process. Your manuscript titled "Interplay of attention-deficit/hyperactivity and problematic online gaming on subsequent mental health issues in adolescents" has now been seen by 3 reviewers, and I include their comments at the end of this message. They find your work of interest but raised some important points. We are interested in the possibility of publishing your study in *Communications Psychology*, but would like to consider your responses to these concerns and assess a revised manuscript before we make a final decision on publication.

We therefore invite you to revise and resubmit your manuscript, along with a point-by-point response to the reviewers. Please highlight all changes in the manuscript text file.

Editorially, we consider it crucial that the reviewers' concerns regarding the validity of the problematic online gaming measures are thoroughly addressed in the revised manuscript. In addition, the advance of the current study in comparison to existing longitudinal studies on ADHD and problematic online gaming should be clearly demonstrated in the revised manuscript.

Response:

Thank you for your consideration. We are delighted to submit the revised version of our manuscript. Below, you will find our detailed responses addressing the reviewers' comments point by point. In response to the editorial request, we have thoroughly addressed two key concerns raised by the reviewers. First, regarding the validity of the problematic online gaming measure, we have clarified that the measure was based on modified gambling disorder criteria rather than the DSM-5 Internet Gaming Disorder framework. We have discussed its conceptual limitations in both the Methods and Discussion sections, newly reported its internal consistency, and acknowledged that its psychometric properties may limit interpretability. Second, we have strengthened the literature review in both the Introduction and Discussion to explicitly introduce prior longitudinal studies on ADHD and problematic online gaming. We have cited recent studies and clearly articulated how our study advances the field by establishing a longitudinal mediation pathway from ADHD symptoms to mental health outcomes via problematic online gaming while applying rigorous causal inference methodology.

Reviewer #1 (Remarks to the Author):

Dear Zui C Narita (and co-authors),

I found your paper titled as "Interplay of attention-deficit/hyperactivity and problematic online gaming on subsequent mental health issues in adolescents" very fascinating and timely. This paper aimed at understanding the potential causes behind the mental health deterioration of adolescents by demonstrating the process of how the presence of ADHD symptoms are leading to common negative psychological outcomes. Regarding the high quality of this paper not only its relevance, but additionally its high methodological rigor is also noteworthy. In the research paper, the authors raise several common methodological issues of previous studies which limit the possibility of establishing more reliable causal inferences (e.g. inappropriate designs or lack of proper handling of covariates), all which they successfully manage. The paper itself is clearly written and includes a well detailed and transparent description of their methodology, results, limitations, practical implications and even directing further research. Besides supporting the publication of the paper, I do have some suggestions which could

potentially improve the quality, mostly (but not only) concerning its theoretical underpinnings and implications.

Response:

We sincerely thank the reviewer for their positive and encouraging feedback. We appreciate the recognition of our work's relevance, methodological rigor, and clarity, and have carefully considered the suggestions to further strengthen the paper.

1. Introduction

The author's statement: "Attention deficit/hyperactivity is often refractory to pharmacotherapy and thus unmodifiable or only partially modifiable."

Comment: The authors provide only one supporting evidence for this general statement. The supporting evidence strictly focuses on aggressive behavior, which does not cover the care needs of ADHD affected youth and is not identical to ADHD symptoms. I recommend mentioning studies where the core symptoms are targeted (several systematic reviews and meta-analyses are already published in this topic). This sentence also gives a bit of a misleading impression that no effective alternatives (such as computerized training) are available currently.

Response:

We thank the reviewer for this comment. Multiple reviewers provided valuable input regarding this description, and we fully agree with the points raised. In response, we have incorporated all comments as follows: we removed the original statement from the Introduction, in line with Reviewer 1's suggestion, and softened the wording in the Discussion section to reflect a more balanced and accurate interpretation, as recommended by Reviewers 1 and 2. We have also cited the publications recommended by the reviewers to support the revised statement. The updated sentence in the Discussion now reads:

"Pharmacological treatment for attention-deficit/hyperactivity is generally effective and widely used²⁰, but it does not necessarily normalize symptoms²¹, i.e., the symptoms could remain only partially modifiable." (page 10, line 20)

The author's statement: "In contrast, POG is a modifiable mediator, and demonstrating its mediation effect could point to potential interventions to mitigate the impact of attention deficit/hyperactivity on mental health issues."

Comment: The authors do not provide any supporting evidence for POG being modifiable. Furthermore, the authors also do not provide evidence for POG being more effectively treated compared to ADHD.

Response:

We agree with the reviewer's concern. As highlighted in the previous response, we have removed these sentences.

The author's statement: "The choice of psychotic experiences, along with depressive symptoms, anxiety, and well-being, was motivated by the fact that while some individuals may hold prejudice against psychotic it is precisely for this reason that we impartially evaluated this common condition during adolescence."

Comment: I do not understand this argument. The inclusion of any expected outcome should be based at least on previous comorbidity studies. Additionally, the paper would greatly

benefit from describing the theorized process of how POG could result in different negative mental health outcomes (either in the introduction or in the discussion).

Response:

We appreciate the reviewer's comment. We have removed the unclear statement regarding the rationale for including psychotic experiences. The inclusion of outcomes is based on established comorbidity with attention-deficit/hyperactivity, as stated in the first paragraph of the Introduction of the original manuscript. In addition, we have added a sentence to clarify how POG could plausibly result in diverse mental health outcomes.

“Given the heterogeneous nature of POG, it may, similar to other forms of addictive behaviors⁸, be associated with a range of negative mental health outcomes depending on individual vulnerabilities and contextual factors.” (page 4, line 3)

General comment for the introduction: The introduction is missing almost (if not all) longitudinal studies focused on the prospective relationship between ADHD and POG or POG and its negative consequences on mental health. As this study is partially aimed at determining a temporal order among these variables, what is already known about the direction of effect must be presented in the introduction.

Response:

We agree with the reviewer's comment. In response, we have added a sentence to the Introduction summarizing previous longitudinal studies showing that attention-deficit/hyperactivity was associated with POG, and that POG, in turn, was associated with negative mental health outcomes.

“Indeed, previous studies suggested the association between attention-deficit/hyperactivity and POG¹⁶⁻¹⁹, as well as between POG and negative mental health outcomes²⁰⁻²³.” (page 4, line 9)

2. Results

The author's statement: “At age 14, boys had a higher mean POG score compared to girls (1.98 vs. 1.52), while at age 16, boys had lower mean scores for depressive symptoms (2.72 vs. 4.85), anxiety (2.26 vs. 2.96), and psychotic experiences (0.24 vs. 0.27), as well as a higher mean score for well-being (67.4 vs. 65.5).”

“At age 12, boys had a higher mean attention-deficit/hyperactivity score compared to girls (3.12 vs. 2.26).”

Comment: When means are reported in your study, please also report standard deviations. Also please be cautious with statements regarding group differences without inferential statistics. Despite most of the differences described are seemingly evident based on the apparent mean differences, others should be referred to as being “close to equal” or “similar”.

Response:

In accordance with the reviewer's suggestion, we have reported not only the gender-specific means but also the corresponding standard deviations in the new Supplementary Table 1. Regarding group differences, we have toned down the wording to avoid implying statistical significance and added clarifying statements to indicate when characteristics appeared similar.

“Gender-specific scores for POG at age 14 and mental health outcomes at age 16 are shown in Supplementary Table 1.” (page 6, line 8)

“Gender-specific scores for attention-deficit/hyperactivity at age 12 are shown in Supplementary Table 1.” (page 8, line 6)

“Participants missing attention-deficit/hyperactivity data appeared more likely to be male, had a higher degree of depressive symptoms and psychotic experiences, a lower degree of anxiety and well-being, and a higher degree of problematic internet use (PIU). Other characteristics appeared similar.” (page 5, line 25)

The author’s statement: “Figure 4 and Table 4 summarize the results for mediation effects. Significant total indirect effects and reasonable proportion mediated were observed across all mental health outcomes (depressive symptoms: 29.2%; anxiety: 12.3%; psychotic experiences: 20.6%; diminished well-being: 22.1%).”

Comment: Meanwhile reporting the proportions mediated is valuable, I would recommend at least briefly reflecting on the differences in predicting your outcomes. The models were clearly more effective in the prediction of well-being of the respondents, compared to having psychotic experiences.

Response:

Thank you for your helpful comment. We acknowledge that in the original manuscript, we mistakenly referred to the regression coefficients as “standardized beta.” In fact, the reported estimates represent β per one SD of the exposure. This has been corrected throughout the revised manuscript. We would also like to clarify that β per one SD does not indicate predictive performance. Its magnitude is influenced by the scale and distribution of the outcome variable, and therefore does not allow for direct comparison across outcomes. For example, psychotic experiences were measured on a narrow scale (0–7), whereas well-being was measured on a broader scale (0–100), which likely contributes to the differences in estimated β values. We have added clarification in the revised manuscript to address this point.

“ β is reported per one SD increase in the exposure; its magnitude depends on each outcome’s scale and is not comparable across outcomes.” (page 20, line 15)

3. Discussion

The author’s statement: “While the reason for the gender-specific results on depression is beyond the scope of this paper, the types of online activities and patterns of engagement may differ between girls and boys.”

Comment: I recommend briefly discussing these findings. It would be beneficial to see if this effect is expected to be generalizable to other cultures, or it may be dependent only on Japanese population. Answering some questions would help others understand if these findings could be population-specific:

- Are females similarly engaged in gaming in Japan as males?
- What is the form of gaming in Japan (e.g. are people tend to play alone on personal consoles/computers potentially leading to isolation from non-virtual social experiences or is it more socially embedded, such as in case of South-Korean PC bangs)?
- Are there gender related differences in game platform or genre preference in Japan?

Response:

We thank the reviewer for these questions. Our dataset does not contain sufficient information to directly examine gender differences in gaming patterns, platforms, or genre preferences.

However, nationally representative survey data from Japan suggest that males tend to spend more time gaming and are more likely to own certain consoles (Egami, et al. 2024). It has also been noted that some consoles are more commonly played alone and others more often in shared, in-person contexts. That said, genre-level preferences by gender were not reported, and we are unable to draw conclusions on this point. We have added a brief contextual note in the Discussion to acknowledge these points.

“While the reason for the gender-specific results on depression is beyond the scope of this paper, Japanese males are reportedly more likely than females to own and play gaming consoles and tend to spend more time playing video games²⁴. Some platforms are more commonly linked to solitary play, whereas others are often used in in-person social contexts²⁴. These differences might be related to heterogeneous patterns of online gaming across genders.” (page 9, line 22)

General comment for the discussion: The discussion is mainly repetition of the results and methods section. Results should only be repeated to clarify if specific results are discussed. Repeating that “...attention-deficit/hyperactivity is often refractory to pharmacotherapy and thus not always modifiable, POG represents a modifiable mediator, highlighting potential intervention points to mitigate its impact on mental health.” is redundant. I would recommend only mentioning this in the discussion, as practical implications are generally expected to be in this section. The main function of the discussion section is the theoretical explanation of results, comparison with other similar findings and reflection on hypotheses and contradictions.

Response:

We agree with the reviewer’s comment. As noted in our earlier responses, we have removed the relevant sentences from the Introduction, as recommended. Also, we have strengthened the Discussion section by adding comparisons with previous studies, which were lacking in the original manuscript.

“These findings are consistent with previous studies showing that POG was associated with later mental health problems²⁰⁻²³.” (page 9, line 17)

“Our findings indicated that attention-deficit/hyperactivity was significantly associated with subsequent POG, consistent with previous studies¹⁶⁻¹⁹” (page 10, line 3)

4. Methods

The author’s statement: Attention-deficit/hyperactivity measurement

Comment: No reliability reported for the subscale used. Was this data self-reported, parent reported or did multiple informants provide information?

Response:

We thank the reviewer for this comment. In response, we have clarified in the Methods section that the attention-deficit/hyperactivity measurement was based on parent reports using the Strengths and Difficulties Questionnaire hyperactivity/inattention subscale. We have also added information on the internal consistency of the subscale.

“To assess attention-deficit/hyperactivity at age 12, we employed the Strengths and Difficulties Questionnaire hyperactivity/inattention subscale³⁵, a parent-report five-item measure widely used in epidemiological studies.” (page 13, line 22)

“The data demonstrated acceptable internal consistency (Cronbach’s α , 0.75).” (page 14, line 3)

The author’s statement: Problematic online gaming measurement

Comment: The authors wrote “We assessed POG at age 14 using a self-reported measure employed in the previous study.” which previous study was not introduced earlier. It is not good practice for referencing measures as it makes it difficult for the potential reader to identify the scale. Authors could simply say “DSM criteria was used” and reference the DSM version where it is originated from. It must be mentioned that it is not the Internet gaming disorder (IGD) criteria, but a modified gambling disorder criteria was used to measure disordered gaming. This measurement has limitations, such as it measures some symptoms which were not incorporated or heavily changed as part of IGD criteria:

- “When not getting what they want in an online game (e.g., they did not get the item they want or did not clear the event), often returning on another day to accomplish this.”

This symptom is seemingly a mixture of “chasing losses” criteria of gambling and the continuation of gaming IGD criteria (but it missed an important aspect, that continuation occurs despite the user experiences negative consequences).

- “Relies on others to provide relief from desperate situations caused by online gaming (e.g., borrowing money).”

Financial problems are evidently more prevalent among problem gamblers (while the overlap between gaming and gambling features is increasingly more visible).

An indication of non-relevance of these symptoms is reflected as both criteria are rarely reported among gamers compared to others (Ide et al., 2021).

This problem is not major and due to IGD criteria first being introduced to DSM in 2013 (APA, 2013), inclusion of this criteria in the current study would have been complicated (considering the necessity of translation, psychometric validation and the submission of a modified ethical permission for the study). But it has to be also mentioned that for dichotomization, gambling criteria cut-off was used (presence of 4 symptoms) instead of IGD criteria cut-off (presence of 5 symptoms). These limitations must be mentioned.

References cited:

Ide, S. et al. Adolescent Problem Gaming and Loot Box Purchasing in Video Games: Cross-sectional Observational Study Using Population-Based Cohort Data. *JMIR Serious Games* 9, e23886 (2021).

American Psychiatric Association. (2013). *Diagnostic and statistical manual of mental disorders: DSM-5*. American psychiatric association.

Response:

We thank the reviewer for these detailed comments. In response, we have clarified in the Methods section that the POG measure was based on a self-reported checklist developed from modified gambling disorder criteria, rather than the proposed internet gaming criteria described in the DSM-5-TR. We have also noted that some items, such as those related to chasing behavior and financial reliance, may reflect constructs characteristic of gambling disorder features. Furthermore, we have clarified that the cut-off for dichotomization (presence of four or more symptoms) was based on gambling disorder standards rather than internet gaming disorder criteria. These clarifications have been incorporated into the revised Methods text.

“We assessed POG at age 14 using a self-reported measure employed in the previous study³⁶. This measure was originally developed based on modified gambling disorder criteria, rather

than the proposed internet gaming disorder criteria in the DSM-5-TR⁸. This measure included nine items with yes/no response options evaluating the presence of behaviors or emotions associated with addiction to online games over the past 12 months. Some items, such as those related to chasing behavior and financial reliance, may reflect constructs characteristic of gambling disorder features. Details of items can be found in Supplementary Table 10. The score was calculated by summing the responses (possible range 0–9), with higher scores indicating greater levels of POG. For dichotomization, a cut-off of four or more symptoms was used, consistent with gambling disorder criteria rather than internet gaming disorder criteria⁸.” (page 14, line 7)

The author’s statement: Mental health outcomes measurement

Comment: What was the timeframe of measurement (past week/month/year)? This is a question for both depression and anxiety measurement. Additionally, for well-being measurement, the 2-week timeframe of measurement is a bit unjustified for a longitudinal study of this length.

Response:

We appreciate this comment. We have added the information regarding the timeframes of each mental health outcome measure as follows in the Methods section. For the latter comment, we struggled to fully understand the reviewer’s intention. If the concern was that the two-week timeframe for the WHO-5 is too short, we respectfully note that our study aimed to capture participants’ mental health status at age 16, rather than over an extended period. If the concern was that the timeframe is too long, we would respectfully note that using a two-week reference period, as seen in widely used measures such as the GAD-7 and PHQ-9, is a standard approach even in cohort studies with multi-year follow-up periods. The revised text now reads:

“To assess depressive symptoms at ages 12 and 16, we employed the SMFQ, a standard 13-item self-report measure^{37,38}, following the recommendation from the NICE guidance³⁹. The scale evaluates depressive symptoms over the past two weeks. Each item had three response options: “always,” “sometimes,” and “never,” scored with 2, 1, and 0 points, respectively. Items were summed into a composite score (possible range 0–26); a greater score suggests a higher degree of depressive symptoms. The data demonstrated strong internal consistency (Cronbach’s α , 0.92).

To assess anxiety at ages 12 and 16, we utilized the Child Behavior Checklist (CBCL)⁴⁰, a widely used caregiver-reported questionnaire consisting of 118 items. To mitigate the burden on participants, we selected 84 items from the CBCL for the survey. Of these, 14 items were specifically designed to assess anxiety, derived from the CBCL anxiety scale (CBCL-A), which initially comprised 16 items. The scale evaluates anxiety symptoms over the past six months. Each item was evaluated on a three-point scale: “not true,” “somewhat or sometimes true,” and “very true or often true.” A total score was calculated by summing the responses for each item, with higher scores indicating a higher degree of anxiety (possible range 0–28). The data demonstrated acceptable internal consistency (Cronbach’s α , 0.78).

To assess psychotic experiences at ages 12 and 16, we used the Adolescent Psychotic-like Symptom Screener (APSS), an established seven-item self-report questionnaire^{41,42}. Each item had three response options: “yes, definitely,” “maybe,” and “no, never,” scored with 1, 0.5, and 0 points, respectively. At age 12, participants were asked about lifetime experiences, while at age 16, they were asked about experiences within the past two years. Items were summed into a total score (possible range 0–7); a greater score suggests a higher degree of psychotic experiences. Details of items can be found in Supplementary Table

11. The data demonstrated acceptable internal consistency (Cronbach's α , 0.71)." (page 14, line 21)

5. Supplementary materials

The author's statement: Supplementary Method 1 | Details on the measurement of covariates.
Comment: the authors dichotomized time spent gaming using the following rule: "less than one hour/day vs. one hour/day or longer" to create a proxy for POG which can be entered as covariate at time 1. This criterion is not justified by earlier research, as an average daily 1-hour videogame use is associated with 0 symptoms of disordered gaming (Pontes et al., 2022, Katz et al., 2024).

Furthermore, while it is understandable that the detailed description of covariate measurement would further lengthen this already long document, I would appreciate if the reliability of the covariate measurement tools would be reported.

References cited:

Pontes, H. M., Schivinski, B., Kannen, C., & Montag, C. (2022). The interplay between time spent gaming and disordered gaming: A large-scale world-wide study. *Social Science & Medicine*, 296, 114721.

Katz, D., Horváth, Z., Pontes, H. M., Koncz, P., Demetrovics, Z., & Király, O. (2024). How much gaming is too much? An analysis based on psychological distress. *Journal of Behavioral Addictions*, 13(3), 716-728.

Response:

We thank the reviewer for this important comment. We agree that one hour of daily video game use alone would not typically indicate disordered gaming. While we considered using a higher threshold for time spent gaming, doing so substantially decreased the number of participants classified above the threshold, leading to concerns about positivity violation. Therefore, we used a lower threshold and combined this variable with problematic internet use to better approximate problematic online gaming at baseline. The potential for residual confounding arising from this approach has been acknowledged in the revised manuscript as a limitation.

"This study has several limitations. First, despite the longitudinal design and the rich confounder adjustment, the possibility of residual confounding remains. Valid inference in causal mediation analysis relies on no-confounding assumptions (see the methods section), but these assumptions are unverifiable. Specifically, we used the set of time spent gaming and PIU as a proxy for POG at age 12; however, it is unverifiable whether adjusting for this set was sufficient to account for the baseline level of POG." (page 11, line 22)

Regarding the latter point, we have added the internal consistency coefficients (Cronbach's α) for the covariate measures that used total scores, as summarized in the revised Supplementary Methods:

"Neighborhood cohesion was assessed using the Neighborhood Collective Efficacy scale⁴, where a higher total score indicates greater neighborhood cohesion (Cronbach's α , 0.87)." (Supplementary Information)

"PIU was evaluated using the modified version of the Compulsive Internet Use Scale, where a higher total score indicates greater PIU (Cronbach's α , 0.87)⁶." (Supplementary Information)

The author's statement: Supplementary Fig. 1 | Association between problematic online gaming at age 14 and mental health at age 16.

Comment: High resolution versions must be inserted, because the figure texts in the current version cannot be read.

Response:

We appreciate this comment. We have replaced Supplementary Figure 1 with a higher-resolution version to improve readability.

The author's statement: Supplementary Table 4 | Sensitivity analysis for the association between problematic online gaming at age 14 and mental health at age 16: analyzing mental health as dichotomous variables.

Comment: naming dichotomous variables in a way that reflects their categorical nature would be better (e.g. instead of "depressive symptoms" using "probable depression", instead of "anxiety" using "high anxiety").

Response:

We thank the reviewer for this comment. In response, we have renamed the dichotomous outcomes in Supplementary Table 4 as "incident depression," "incident anxiety," and "incident psychotic experiences." In addition, we have modified the relevant parts of the main text and other supplementary tables to ensure consistency.

"Significant additive interaction was shown for incident depression (RERI 1.21, 95% CI 0.11–2.55), suggesting the absolute risk increase was larger in girls (Supplementary Table 2)." (page 7, line 8)

"Third, we examined the E-values for the doubly robust estimator and found that substantial unmeasured confounding would be required to explain away the observed estimates (E-values for point estimates: 2.62 for incident depression, 3.37 for incident anxiety, 2.82 for incident psychotic experiences, and 2.45 for diminished well-being) (Supplementary Table 5)." (page 7, line 18)

General comment for the whole paper: Disordered gaming is referred to as "POG engagement", which is a strange choice of wording. I would recommend referring to it as "severity", as a clinical disorder was being measured, not the intensity or frequency of participation in a recreational activity (as it would be if time spent playing games would have been measured, as an outcome).

Response:

We thank the reviewer for this insightful suggestion. After careful consideration, we recognized that the term "POG engagement" was not optimal, particularly in the context of clinical measurement. In the revised manuscript, we have replaced it with "high degree of POG" and "low degree of POG" to more accurately reflect the binary categorization used in our analyses. While we agree that "severity" would be a more appropriate term for a continuous measure, we opted for terminology that aligns with the dichotomous nature of our variable.

I wish you good luck with publishing your results.

Patrik Koncz
Research Group Member, Lecturer
Addiction Research Group
Eötvös Loránd University, Faculty of Education and Psychology

Response:

We sincerely thank Dr. Koncz for the thoughtful feedback and kind wishes.

Reviewer #2 (Remarks to the Author):

Thank you for the opportunity to review the submission of the journal article “Interplay of attention-deficit/hyperactivity and problematic online gaming on subsequent mental health issues in adolescents” (COMMSPSYCHOL-25-0086-T) in Communications Psychology. This article poses an interesting and clinically relevant questions about the relationship between ADHD and mental health concerns and its mediation by problematic online gaming. The article is well aligned with the journal’s scope of publishing brief yet impactful research studies and could be published after minor revisions.

The authors used existing data to conduct a robust mediation analysis and took additional steps to ensure the robustness of the analysis. The outcomes of their analysis are clear, and likely to be applicable to similar populations, making it worthwhile to share this analysis with the readers of Communications Psychology. Recommendations to improve the article were for the most part fairly conceptual, and the authors may choose to enact these as they see fit.

I divide my reviews into key points and minor comments, as most of the minor comments are issues that can be fixed relatively quickly, with major points often requiring further thought.

Response:

We sincerely thank the reviewer for the thoughtful and encouraging feedback. We appreciate the recognition of our work and have carefully addressed all comments raised.

Key Points:

1. Overstatement of treatment-resistant ADHD: Pharmacological treatment of ADHD is actually one of the most effective psychiatric pharmacological interventions available, so while it is true that some cases are treatment-refractory I think stating that this happens “often” is over-stating the claim (Mechler et al., 2022). I don’t think the authors need to claim that ADHD is hard to modify for them to argue that problematic gaming is an important target, as problematic gaming is itself associated with negative outcomes (as the authors highlight). Similarly, psychological therapies which are often helpful adjuncts in the treatment of ADHD are not targeting “core symptoms” of ADHD, but rather the management of the disorder – and I see a similar role for interventions targeting problematic gaming.

Response:

We thank the reviewer for this comment. Multiple reviewers provided valuable input regarding this description, and we fully agree with the points raised. In response, we have incorporated all comments as follows: we removed the original statement from the Introduction, in line with Reviewer 1’s suggestion, and softened the wording in the

Discussion section to reflect a more balanced and accurate interpretation, as recommended by Reviewers 1 and 2. We have also cited the publications recommended by the reviewers to support the revised statement. The updated sentence in the Discussion now reads:

“Pharmacological treatment for attention-deficit/hyperactivity is generally effective and widely used²⁰, but it does not necessarily normalize symptoms²¹, i.e., the symptoms could remain only partially modifiable.” (page 10, line 20)

2. Sample representativeness: It would be good to understand who the sample is meant to represent. For example, was the final sample from the three municipalities selected broadly similar to the population of adolescents in the Greater Tokyo area and therefore likely to represent urban middle-class Japanese adolescents with mild-to-moderate ADHD symptoms (given the high rate of drop-out from those with high ADHD symptoms)?

Response:

We appreciate the reviewer’s comment. We clarify that the sample was drawn from three municipalities (Setagaya, Mitaka, and Chofu) in Japan, all of which are urban areas. All participants, including those who later dropped out, were included in the analyses using random forest imputation for missing data. Therefore, the risk of selection bias due to ADHD symptom severity is likely to be minimal. Regarding representativeness, we have acknowledged in the Discussion section that the sample predominantly consisted of adolescents from a highly urbanized city and largely of Asian ethnicity, which may limit the generalizability of the findings to more rural or racially diverse populations. Specifically, we stated:

“Fourth, the transportability of our findings may be limited; the sample, predominantly of Asian ethnicity from the highly urbanized city, may not represent rural or diverse racial demographics well.” (page 12, line 10)

3. Problematic online gaming measure: Given this is a custom measure further information is warranted before treating the summative score as an adequate summation of the items in the scale. For example, the authors do not report whether the questions load onto a single factor (in either this study or the study cited as the source of the measure – reference 31); likewise, it may be helpful to include a standardised Cronbach alpha value as calculated on the basis of the participants’ responses to the POG measure, as was reported for the mental health outcomes.

Response:

We agree with the reviewer that adding the Cronbach alpha may be helpful. In response, we have calculated and reported the limited internal consistency of the POG measure in the Methods section of the revised manuscript. We have also acknowledged this limitation in the Discussion section:

“The data demonstrated limited internal consistency (Cronbach’s α , 0.64), and thus the results should be interpreted with caution.” (page 14, line 17)

“Moreover, the POG measure demonstrated limited internal consistency, which warrants caution when interpreting the findings.” (page 12, line 6)

4. Considerations regarding gender differences: The authors may want to consider

downplaying gender differences given findings were highly significant across genders, and there are important reporting differences between girls and boys (with girls more likely to report mood symptoms in the first place). At the same time, there is some evidence suggesting greater “addictiveness” in girls when engaging with specific forms of substance use (and possibly other behavioural addictions). In my view this would amount to more tentative language regarding the gender-based differences.

Response:

We appreciate the reviewer’s thoughtful comment. We respectfully note that the larger absolute risk increase for depression in girls, as indicated by the relative excess risk due to interaction, reflects an additive increase beyond the baseline risk. Therefore, while girls are generally more likely to report mood symptoms, the additional risk associated with POG is still larger in girls. That said, we agree with the reviewer that it would be more appropriate to adopt a conservative tone regarding gender-based differences. Accordingly, we have removed statements such as “the findings were more consistent in girls than boys” to reflect a more cautious interpretation.

Minor Comments:

1. There’s a period missing at the end of aim 3.

Response:

We thank the reviewer for pointing this out. We have added a period at the end of aim 3.

2. Page 5, Line 80: The sentence regarding the “choice of psychotic experiences” seems out of place, as by this point the reader is not aware of what is or is not included in the assessment. Please consider rewording this statement.

Response:

We appreciate the reviewer’s comment. In response, we have removed the sentence from the Introduction to improve the logical flow.

3. The robustness of the statistical analysis is encouraging to see.

Response:

We sincerely thank the reviewer for the positive feedback regarding the robustness of our statistical analysis.

4. Ensure the accessibility of Figures by testing whether chosen colour schemes are able to be accurately perceived by individuals with colour impairments.

Response:

We thank the reviewer for this important comment. In response, we have modified the color schemes used in Figures 1, 2, 3, and 4 to enhance accessibility for individuals with color vision impairments. We adopted a color combination that is widely recognized for being distinguishable under common types of color impairments.

Reviewer #3 (Remarks to the Author):

The study entitled “Interplay of attention-deficit/hyperactivity and problematic online gaming on subsequent mental health issues in adolescents” had some strengths, including a

representative and relatively large sample (n=3171), a longitudinal design with four timepoints, several different standardized instruments assessing mental health (including SMFQ, CBCL, APSS, and WHO-5), and rigorous data analyses. The authors found that problematic online gaming is a significant mediator in the longitudinal relationships between ADHD and mental health issues. Although I appreciate the authors' findings and above-mentioned strengths, I have some concerns in the present study. Below please see my specific comments.

Response:

We are grateful to the reviewer for the careful review and constructive feedback, which have greatly contributed to strengthening our manuscript. Below, you will find our detailed responses addressing your comments point by point.

1. The authors claimed that “However, evidence specifically examining the impact of problematic online gaming (POG) on mental health issues remains limited”; however, this is inaccurate. There is ample evidence showing the relationship between problematic online gaming and mental health issues, including population with ADHD and some longitudinal studies on other populations. Below are some references I know (only on ADHD and longitudinal studies); however, I believe that the authors could find a lot in the databases, especially using cross-sectional design.

On ADHD:

Chen, C.-Y., Lee, K.-Y., Fung, X. C. C., Chen, J.-K., Lai, Y.-C., Potenza, M. N., Chang, K.-C., Fang C.-Y., Pakpour, A. H., & Lin, C.-Y. (2024). Problematic use of internet associates with poor quality of life via psychological distress in individuals with ADHD. *Psychology Research and Behavior Management*, 17, 443-455.

Lee, K.-Y., Chen, C.-Y., Chen, J.-K., Liu, C.-C., Chang, K.-C., Fung, X. C. C., Chen, J.-S., Kao, Y.-C., Potenza, M. N., Pakpour, A. H., Lin, C.-Y. (2023). Exploring mediational roles for self-stigma in associations between types of problematic use of internet and psychological distress in youth with ADHD. *Research in Developmental Disabilities*, 133, 104410.

Using longitudinal design:

Chang, C.-W., Huang, R.-Y., Strong, C., Lin, Y.-C., Tsai, M.-C., Chen, I.-H., Lin, C.-Y., Pakpour, A. H., & Griffiths, M. D. (2022). Reciprocal relationships between problematic social media use, problematic gaming, and psychological distress among university students: A nine-month longitudinal study. *Frontiers in Public Health*, 10, 858482.

Chen, I.-H., Lin, Y.-C., Lin, C.-Y., Wang, W.-C., & Gamble, J. H. (2022). The trajectory of psychological distress and problematic Internet gaming among primary school boys: a longitudinal study across different periods of COVID-19 in China. *Journal of Men's Health*, 18(3), 070.

Chen, C.-Y., Chen, I.-H., Hou, W.-L., Potenza, M. N., O'Brien, K. S., Lin, C.-Y., & Latner, J. D. (2022). The relationship between children's problematic Internet-related behaviors and psychological distress during the onset of the COVID-19 pandemic: A longitudinal study. *Journal of Addiction Medicine*, 16, e73-e80.

Chen, I.-H., Chen, C.-Y., Pakpour, A. H., Griffiths, M. D., Lin, C.-Y., Li, X.-D., Tsang, H. W. H. (2021). Problematic internet-related behaviors mediate the associations between levels of internet engagement and distress among schoolchildren during COVID-19 lockdown: A longitudinal structural equation modeling study. *Journal of Behavioral Addictions*, 10(1), 135-148.

Response:

We agree with the reviewer that previous research examining the association between POG and mental health issues should be cited. This point was also raised by other reviewers. In the revised manuscript, we have removed the statement that “However, evidence specifically examining the impact of problematic online gaming (POG) on mental health issues remains limited.” We have cited previous longitudinal studies, as suggested by Reviewer 1. In addition, we appreciate Reviewer 3’s helpful references, several of which have now been cited in the revised Introduction:

“Indeed, previous studies suggested the association between attention-deficit/hyperactivity and POG¹⁶⁻¹⁹, as well as between POG and negative mental health outcomes²⁰⁻²³.” (page 4, line 9)

2. I disagree with the statement “For POG, we used symptom-based measures to differentiate it from online gaming in general, thereby addressing the limitations of past research”. Many studies have used DSM-5-based measure (e.g., Internet Gaming Disorder Scale) to examine the relationship between problematic gaming and mental health issues.

Response:

We agree with the reviewer’s comment and have accordingly removed the sentences.

3. Lines 89-92, please use citation to support the key considerations.

Response:

We have added a citation as the reviewer suggested.

4. The statement “Unfortunately, even in leading behavioral science journals, studies employing mediation analysis often encounter these methodological challenges” is vague as (i) it is unclear what “leading behavioral science journals” are and (ii) there are no citations to support this statement.

Response:

We agree with the reviewer that the original sentence was unclear. In response, we have removed the phrase “even in leading behavioral science journals” and have added a citation to support the revised statement. The revised sentence now reads:

“Unfortunately, studies employing mediation analysis often encounter these methodological challenges, which hinder their interpretations and the applicability of their findings to real-world contexts²⁴.” (page 5, line 9)

5. The authors used the abbreviation of PIU but did not define it as problematic internet use for its first-time use.

Response:

We thank the reviewer for this comment. In response, we have defined PIU as "problematic internet use" at its first occurrence in the revised manuscript.

6. Following the previous comment, it is unclear how the authors collected PIU (i.e., what measure was used).

Response:

Information regarding the measurement of PIU, including the scale used and the relevant citation, was already provided in the Supplementary Information of the original manuscript. In response to another reviewer's suggestion, we have additionally added the internal consistency (Cronbach's $\alpha = 0.87$) for this measure in the revised Supplementary Information.

"PIU was evaluated using the modified version of the Compulsive Internet Use Scale, where a higher total score indicates greater PIU (Cronbach's $\alpha, 0.87$)⁶." (Supplementary Information)

7. Results section repeats some information from Methods.

Response:

We thank the reviewer for this comment. As this journal follows the Nature Portfolio format where the Results section precedes the Methods section, we included brief reminders of the analytic approaches in the Results to facilitate readers' understanding. However, in response to the reviewer's suggestion, we have removed the detailed description of the doubly robust estimation procedure (e.g., inverse probability weighting, generalized linear models, g-formula) from the Results section to avoid redundancy. Readers are now referred to the Methods section for these details.

8. I suppose that the authors used the 9 criteria defined by the DSM-5 to assess POG. However, this is not clearly stated.

Response:

We appreciate this comment. Multiple reviewers commented on this issue, and we have thoroughly updated the explanations for this measure:

"We assessed POG at age 14 using a self-reported measure employed in the previous study³⁹. This measure was originally developed based on modified gambling disorder criteria, rather than the proposed internet gaming disorder criteria in the DSM-5-TR⁸. This measure included nine items with yes/no response options evaluating the presence of behaviors or emotions associated with addiction to online games over the past 12 months. Some items, such as those related to chasing behavior and financial reliance, may reflect constructs characteristic of gambling disorder features. Details of items can be found in Supplementary Table 10. The score was calculated by summing the responses (possible range 0–9), with higher scores indicating greater levels of POG. For dichotomization, a cut-off of four or more symptoms was used, consistent with gambling disorder criteria rather than internet gaming disorder criteria⁸. The data demonstrated limited internal consistency (Cronbach's $\alpha, 0.64$), and thus the results should be interpreted with caution." (page 14, line 7)

9. It is unclear where the participants complete the questionnaires, and how the procedure was done.

Response:

We appreciate the reviewer's comment. The procedure for administering the questionnaires was already described in the original manuscript. In response to the reviewer's suggestion, we have clarified the location by adding that trained interviewers visited the participants' homes to administer the self-report questionnaires.

"Trained interviewers visited the participants' homes and provided self-report questionnaires to child-parent pairs at each time point." (page 13, line 12)

10. It is unclear if all the standardized measures have been translated into Japanese with validity evidence.

Response:

We agree with the reviewer that this issue should be considered. We acknowledge that the standardized measures used in the study were not formally validated in Japanese. In response, we have enriched the second limitation in the Discussion section to acknowledge this issue.

“Second, the potential for measurement bias exists, particularly with self-reported measures, which might not accurately capture the actual pathology as a clinical interview would. Also, the standardized measures used in this study were not formally validated in Japanese, which could further contribute to measurement bias.” (page 12, line 2)

11. It is unclear how the authors assessed some covariates (e.g., IQ, loneliness, physical punishment neighborhood cohesion, and gender nonconforming behavior).

Response:

Again, the information for covariates measures was already provided in the Supplementary Information of the original manuscript. Only the information for IQ was not shown in the original manuscript, and thus we have added the relevant sentence.

“IQ was evaluated using the short form of the Wechsler Intelligence Scale for Children-Third Edition¹.” (Supplementary Information)

12. I would suggest using years instead of months to present age. If they are toddlers or young children, using months is better. However, they are already over 10 years and interpreting age using months for this population is non-intuitive.

Response:

As suggested by the reviewer, we have revised the presentation of participants' age from months to years throughout the manuscript to improve readability.

13. I would suggest deleting some overlapped description in Results section. That is, if the information is not that important and can be read in Tables, the authors need not to mention the values/statistics in Results section.

Response:

We agree with the reviewer that some of the information was redundant. Accordingly, we have removed descriptions in the Results section that overlapped with the information presented in the Tables.

14. The inclusion of sensitivity analyses is a strength.

Response:

We sincerely thank the reviewer for the positive feedback regarding the robustness of our sensitivity analysis.

15. I believe that the authors have to search the literature again to strengthen their Discussion

section (i.e., using the literature on the relationship between problematic gaming and mental health issues to communicate with the present findings).

Response:

Thank you for your helpful comment. In accordance with your suggestion, we have incorporated references to previous longitudinal studies into the Discussion section. Specifically, we added a sentence highlighting that our findings on the association between POG and subsequent mental health problems are consistent with prior research. Also, we included a sentence noting that the association between attention-deficit/hyperactivity and subsequent POG aligns with previous studies.

“These findings are consistent with previous studies showing that POG was associated with later mental health problems²⁰⁻²³.” (page 9, line 17)

“Our findings indicated that attention-deficit/hyperactivity was significantly associated with subsequent POG, consistent with previous studies¹⁶⁻¹⁹” (page 10, line 3)

Dear Dr Narita,

Your manuscript titled "Problematic online gaming mediates the association between attention-deficit/hyperactivity and subsequent mental health issues in adolescents" has now been seen by our reviewers, whose comments appear below. In light of their advice I am delighted to say that we are happy, in principle, to publish a suitably revised version in *Communications Psychology*.

We therefore invite you to revise your paper one last time to address the remaining concerns of our reviewers and a list of editorial requests. At the same time we ask that you edit your manuscript to comply with our format requirements and to maximise the accessibility and therefore the impact of your work.

EDITORIAL REQUESTS:

Response:

Thank you for your consideration and the helpful feedback. We have revised the manuscript to address the final reviewer concern.

In addition, we have addressed all remaining editorial requests as outlined in the Editorial Requests Table. Specifically, we made the following revisions:

1. **Manuscript Structure:** The Methods section has been relocated to follow the Introduction, as required.
2. **Terminology:** All instances of the abbreviation "POG" were replaced with "problematic online gaming." We also reviewed the manuscript to ensure appropriate terminology regarding gender and sex; we use "girls" and "boys" or "women" and "men" and avoided using "male" and "female" as nouns.
3. **Assumption Statement:** We added a statement clarifying that normality of the data was assumed for regression analyses involving continuous outcomes, although this assumption was not formally tested.
4. **Discussion:** We revised the Discussion and Abstract to remove any suggestions of policy or clinical interventions. We now refer only to "further intervention research" without implying direct application. Moreover, we have removed a whole paragraph related to policy recommendations.
5. **Statistical Reporting:** We now report exact p-values unless $p < 0.001$ and have included full statistical details throughout the text.
6. **Figure Updates:** We added the number of participants to Figure 2, as requested.
7. **Data Availability:** The numerical data underlying Figure 2 have been deposited in Zenodo, and the link is included in the revised Data Availability section and References.
8. **Code Availability:** The GitHub repository has been archived in Zenodo and cited with its DOI in both the Code Availability section and the References.

We have documented all changes in the completed Editorial Requests Table, which has been uploaded as a Related Manuscript file.

We hope the revised version is now suitable for publication in *Communications Psychology*.

Reviewer #1 (Remarks to the Author):

Dear Zui C Narita (and co-authors),

Your paper has developed significantly thanks to the revision. I do appreciate your transparency regarding the validity of POG measurement, I think in the revised version the manuscript fairly reflects on this limitation. Furthermore, I still have one minor concern regarding one of the applied solutions, but besides that, I recommend your paper for publication. In my response I copied the earlier discussion first (your “Earlier author’s statement”, my “Earlier reviewer comment”), then your response to the comments and my new comments at the end.

Earlier author’s statement: “In contrast, POG is a modifiable mediator, and demonstrating its mediation effect could point to potential interventions to mitigate the impact of attention deficit/hyperactivity on mental health issues.”

Earlier reviewer comment: The authors do not provide any supporting evidence for POG being modifiable. Furthermore, the authors also do not provide evidence for POG being more effectively treated compared to ADHD.

Response: We agree with the reviewer’s concern. As highlighted in the previous response, we have removed these sentences.

New comment: I do think that POG is modifiable (at least on short-term) (see Stevens et al., 2019), the problem was not supporting it by a proper study.

Reference cited: Stevens, M. W., King, D. L., Dorstyn, D., & Delfabbro, P. H. (2019).

Cognitive-behavioral therapy for Internet gaming disorder: A systematic review and meta-analysis. *Clinical psychology & psychotherapy*, 26(2), 191-203.

Wishing you good luck with the publication of your results,

Patrik Koncz

Research Group Member, Lecturer

Addiction Research Group

Eötvös Loránd University, Faculty of Education and Psychology

Response:

We appreciate the reviewer for providing the relevant reference. We have accordingly cited the study in the revised manuscript:

“In such cases, problematic online gaming may represent a modifiable mediator⁵⁷. Further intervention study is warranted to examine its potential as a treatment target.”

57. Stevens, M. W. R., King, D. L., Dorstyn, D. & Delfabbro, P. H. Cognitive-behavioral therapy for Internet gaming disorder: A systematic review and meta-analysis. *Clin. Psychol. Psychother.* 26, 191–203 (2019).

Reviewer #2 (Remarks to the Author):

Thank you for the chance to review the authors' responses to my comments.

I am satisfied that my comments have been addressed carefully and successfully.

Response:

We are grateful for the reviewer's positive feedback and thank them for their time and effort in helping us improve the manuscript.

Reviewer #3 (Remarks to the Author):

The authors have satisfactorily addressed my previous comments. I am happy with the revision and the presentation now looks good to me.

Response:

We thank the reviewer for their constructive comments throughout the review process. We are delighted that they are satisfied with the revised manuscript.